# Blind deconvolution estimation by multi-exponential models and alternated least squares approximations: Free-form and sparse approach

**Daniel U. Campos-Delgado**[1,2]*, **Omar Gutierrez-Navarro**[3], **Ricardo Salinas-Martinez**[1], **Elvis Duran**[4], **Aldo R. Mejia-Rodriguez**[1], **Miguel J. Velazquez-Duran**[1], **Javier A. Jo**[5]

**1** Facultad de Ciencias, Universidad Autonoma de San Luis Potosi, San Luis Potosi, Mexico, **2** Instituto de Investigacion en Comunicacion Optica, Universidad Autonoma de San Luis Potosi, San Luis Potosi, Mexico, **3** Department of Biomedical Engineering, Universidad Autonoma de Aguascalientes, Aguascalientes, Mexico, **4** Department of Biomedical Engineering, Texas A&M University, College Station, Texas, United States of America, **5** School of Electrical and Computer Engineering, University of Oklahoma, Norman, Oklahoma, United States of America

* ducd@fciencias.uaslp.mx

**Data Availability Statement:** All the software implementations of the studied methodologies are

## Abstract

The deconvolution process is a key step for quantitative evaluation of fluorescence lifetime imaging microscopy (FLIM) samples. By this process, the fluorescence impulse responses (FluoIRs) of the sample are decoupled from the instrument response (InstR). In blind deconvolution estimation (BDE), the FluoIRs and InstR are jointly extracted from a dataset with minimal a priori information. In this work, two BDE algorithms are introduced based on linear combinations of multi-exponential functions to model each FluoIR in the sample. For both schemes, the InstR is assumed with a free-form and a sparse structure. The local perspective of the BDE methodology assumes that the characteristic parameters of the exponential functions (time constants and scaling coefficients) are estimated based on a single spatial point of the dataset. On the other hand, the same exponential functions are used in the whole dataset in the global perspective, and just the scaling coefficients are updated for each spatial point. A least squares formulation is considered for both BDE algorithms. To overcome the nonlinear interaction in the decision variables, an alternating least squares (ALS) methodology iteratively solves both estimation problems based on non-negative and constrained optimizations. The validation stage considered first synthetic datasets at different noise types and levels, and a comparison with the standard deconvolution techniques with a multi-exponential model for FLIM measurements, as well as, with two BDE methodologies in the state of the art: Laguerre basis, and exponentials library. For the experimental evaluation, fluorescent dyes and oral tissue samples were considered. Our results show that local and global perspectives are consistent with the standard deconvolution techniques, and they reached the fastest convergence responses among the BDE algorithms with the best compromise in FluoIRs and InstR estimation errors.

freely available in the website: http://galia.fc.uaslp.mx/~bde.

**Funding:** This study was supported by the National Institute of Health (NIH) (R01CA218739), Cancer Prevention and Research Institute of Texas (CPRIT grant RP180588), and by CONACYT through a Basic Research Grant (Ref # 254637).

**Competing interests:** All authors have declared that no competing interests exist.

## Introduction

Fluorescence microscopy has become a powerful tool to characterize the chemical properties of tissue samples [1–4]. In fluorescence lifetime imaging microscopy (FLIM), a sample is excited by an electromagnetic source, usually an ultraviolet (UV) narrow laser-pulse. Given a sample with synthetic dyes or endogenous fluorophores, the sample will emit light at a longer wavelength than the excitation source [3, 4]. This fluorescent response contains information of the chemical environment in the sample, its fluorescent components and their concentrations. In multi-spectral FLIM (mFLIM) several spectral channels are recorded simultaneously, and the resulting time-responses are concatenated to construct a characteristic time-frequency signature [5, 6]. The mFLIM methodology has shown to be a powerful medical resource for early and non-invasive diagnosis for different pathologies such as cardiovascular and dermatological diseases [5, 7–9], oral pre-cancer and cancer conditions [10, 11], colonic dysplasia [12], skin cancer [13, 14], and to assess therapeutic responses of anticancer drugs [15].

The fluorescence response measured by FLIM can be modeled as the convolution between the instrument response (InstR) and the particular fluorescence impulse response (FluoIR) of the tissue sample. In order to identify the FluoIR of the sample and provide quantitative information of the FLIM data, a deconvolution stage needs to isolate the InstR from the fluorescence decay (FluoD) [16–20]. There are different strategies to solve this inverse problem, usually the InstR is assumed known or measured a priori, and then carefully aligned with the FluoIRs to avoid bias in the estimations. Other strategies quantify FLIM data by analyzing the FluoDs with a linear unmixing approach [21–25], or in a lower-dimensional domain using the phasor approach [26–28].

In the literature, there are two principal tendencies for the deconvolution process for FLIM datasets: 1) multi-exponential models [16–18], and 2) Laguerre-basis approximation [19, 20, 29]. The former assumes that the FluoIR of each fluorophore in the sample is a linear combination of multi-exponential functions, characterized by the time constants for the exponential functions and the scaling coefficients for each sample position (*local approach*). On the other hand, a *global approach* on the time constants is defined when these parameters are assumed constant throughout the sample, i.e. just the scaling coefficients of the exponential functions change for every sample position. Given the non-linear dependence on the time constants, a non-linear least squares approach is assumed to estimated these parameters [30], while the scaling coefficients can be solved using standard least squares routines for each spatial point. In other approaches, the FluoIR is modeled as a linear combination of discrete-time Laguerre basis functions. In this way, the objective is to estimate the scaling coefficients of the Laguerre functions, whose values can be computed through a linear least squares estimation. The disadvantage of this approach is that for some cases, the resulting FluoIRs might not have a monotonic decay, which is a characteristic of the fluorescence decays, or the fitting could be poor for extremely short or long lifetimes [4]. Therefore, a constrained optimization method is applied with restrictions on the second or third order time derivatives of the resulting FluoIRs to overcome this drawback. Finally, fluorescence lifetimes can be computed from the estimated FluoIRs to provide information about the chemical composition of the sample [31].

An alternative approach is blind deconvolution estimation (BDE), which aims to simultaneously estimate the InstR and the FluoIRs in the sample [32]. Under this perspective, the resulting InstR will be automatically aligned with the FluoDs to minimize the bias estimation of the FluoIRs. To the best of the author's knowledge, there is only one BDE algorithm reported for FLIM data, which considers a Laguerre-basis approach [32]. However, a key hyper-parameter of this proposal has to be carefully tuned for the given input data, and it requires a constrained optimization scheme to achieve a monotonically decaying FluoIR. An

equivalent formulation to BDE was presented in [33] for spike train inference from fluorescence measurements to evaluate neurons activity. In this formulation, the generative model is linear auto-regressive, and the spike inference is obtained by an approximated maximum a posterior formulation with a sparse constraint. This same formulation is further extended in [34] by an active set method to solve the sparse non-negative deconvolution problem (non-negative least absolute shrinkage and selection operator, LASSO, formulation) motivated by isotonic regression. Recently, in [35], the authors introduced the short-and-sparse deconvolution methodology which is based on a bilinear LASSO problem, sphere constraints and data-driven initialization.

In this context, this paper introduces two new BDE algorithms based on modeling the FluoIR by a linear combination of multi-exponential functions. The first BDE algorithm seeks for the characteristic parameters of the exponential functions (time constants and scaling coefficients) with a local perspective in each spatial point of the sample, i.e. pixel-by-pixel. On the other hand, in the global approach, the exponential functions in the FluoIR model are assumed common for all spatial points of the dataset, while their contributions change across the sample, i.e. the time constants of the exponentials are common to all pixels, but the scaling coefficients in each pixel are different. Given the monotonically decaying nature of the exponential functions, there is no need to include a shape-constraint during the estimation process in contrast to [32]. In addition, the only hyperparameters are the lower and upper bounds for the time constants of the exponential functions, which can be easily defined based on prior information of the fluorophores expected in the sample. To overcome the nonlinear interaction between FluoIRs and InstR variables, an alternating least squares (ALS) methodology, iteratively solves both estimation problems. In fact, due to a linear dependence on the InstR parameters in the observation model, a global search is made to estimate and align the shape of the UV laser-pulse through a linear non-negative least squares approximation. The InstR estimation is assumed with a free-form structure and a sparcity condition. In this way, our BDE methods jointly provides an estimation of the InstR and FluoIRs in the sample. The synthetic and experimental results, including synthetic datasets, fluorescence dyes, and oral tissue samples, show that the proposals are robust to different noise types and levels, and achieve a low computational time compared to other strategies in the state of the art. An initial version of the BDE algorithm based on the global approach was presented in [36]. Contrary to [33, 34], in our approach, the InstR does not have a spike train shape, and the observation model relies on a convolution with a multi-exponential structure. Furthermore, our formulation does not assume a short kernel nor a sparse activation map, as in comparison to [35]; and also our multi-exponential kernel is not restricted to a sphere constraint.

The notation used in this paper is described next. Scalars are denoted by italic letters, while vectors and matrices by lower and upper-case bold letters, respectively. $\mathbb{R}$ and $\mathbb{Z}$ represent the real and integer numbers, respectively, and $\mathbb{R}^N$ $N$-dimensional real vectors. For a real vector $\mathbf{x}$, the transpose operation is denoted by $\mathbf{x}^\top$, the $l$-th element by $\mathbf{x}[l]$, the Euclidean norm by $\|\mathbf{x}\|_2 = \sqrt{\mathbf{x}^\top \mathbf{x}}$, and $\mathbf{x} \geq 0$ ($\mathbf{x} > 0$) represents that each element in the vector is non-negative (positive). For a square matrix $\mathbf{X} \in \mathbb{R}^{N \times N}$, $X_{i,j}$ represents the element in the $i$-th row and $j$-th column ($i, j \in [1, N]$), $\mathrm{Tr}(\mathbf{X}) = \Sigma_i X_{i,i}$ denotes the trace operation, and $\|\mathbf{X}\|_F = \sqrt{\mathrm{Tr}(\mathbf{X}^\top \mathbf{X})}$ the Frobenius norm. $\mathbf{X} \geq 0$ represents that each element in the matrix is non-negative. For all vectors $\mathbf{x} \in \mathbb{R}^N$ and $\mathbf{y} \in \mathbb{R}^M$, $T_{\mathbf{x},\mathbf{y}} \in \mathbb{R}^{N \times M}$ denotes a Toeplitz matrix with $\mathbf{x}$ and $\mathbf{y}^\top$ as its first column and row, respectively, where the first element in $\mathbf{x}$ and $\mathbf{y}$ must be equal. An $N$-dimensional vector for which all elements are ones (zeros) is represented by $\mathbf{1}_N$ ($\mathbf{0}_N$), and $\mathbf{I}_N$ denotes the identity matrix of order $N$. For a random variable $x$, $x \sim \mathcal{U}[a, b]$ represents that $x$

is uniformly distributed in the interval $[a, b]$ $(b > a)$, and $x \sim \mathcal{N}(0, \sigma^2)$ that $x$ is normally distributed with zero mean and variance $\sigma^2$.

## Blind deconvolution estimation

We assume that the FLIM dataset is sampled regularly with a period $T$ over a spatial domain of $K$ points in the field of view (FOV) of the instrument [1–3]. Hence, the FluoDs at any spatial location are $L$-th dimensional vectors $\mathbf{z}_k \in \mathbb{R}^L$ that satisfy $\mathbf{z}_k \geq 0 \; \forall k \in [0, K-1]$. The set of FLIM measurements is denoted as $\mathcal{Z} = \{\mathbf{z}_0, \dots, \mathbf{z}_{K-1}\}$, where their spatial location is not relevant so the ordering in $\mathcal{Z}$ is indistinct. Considering an initial pre-processing stage, all the measured FluoDs are normalized to sum-to-one, therefore:

$$\mathbf{y}_k = \frac{\mathbf{z}_k}{\mathbf{1}^\top \mathbf{z}_k} \quad \forall k \in [0, K-1], \tag{1}$$

such that the dataset of scaled FluoDs is given by $\mathcal{Y} = \{\mathbf{y}_0, \dots, \mathbf{y}_{K-1}\}$ and $\mathrm{card}(\mathcal{Y}) = K$. The FLIM observation model for the $l$-th time sample in the $k$-th spatial position is given by

$$\mathbf{y}_k[l] = \mathbf{u}[l] \star \mathbf{h}_k[l] + \mathbf{v}_k[l] = \sum_{j=0}^{l} \mathbf{u}[l-j]\,\mathbf{h}_k[j] + \mathbf{v}_k[l] \quad \forall l \in [0, L-1], \; k \in [0, K-1], \tag{2}$$

where $\mathbf{y}_k[l]$, $\mathbf{u}[l]$ and $\mathbf{h}_k[l]$ denote the measured FluoD, InstR, and FluoIR, respectively, $\star$ represents the convolution operator, and $\mathbf{v}_k[l]$ stands for random noise related to measurement uncertainty. As a result, the scaled FluoDs represent the causal convolution between the InstR and the FluoIR at each spatial location [37]. The FLIM observation model is depicted in Fig 1.

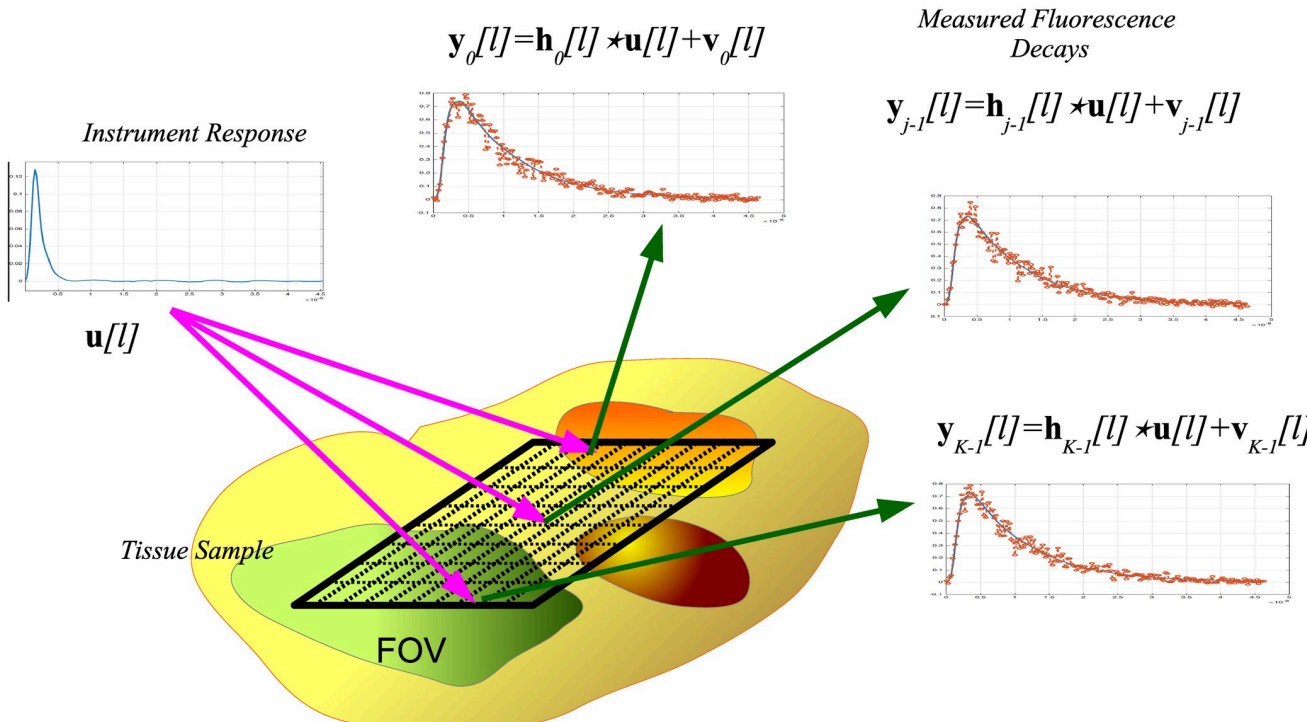

**Fig 1. FLIM observation model.** The instrument response $\mathbf{u}[l]$ is common to all $K$ spatial points, and the FluoDs $\{\mathbf{y}_k[l]\}_{k=0}^{K-1}$ result from the convolution between the FluoIRs $\{\mathbf{h}_k[l]\}_{k=0}^{K-1}$ and the InstR $\mathbf{u}[l]$.

In our formulation, the FluoIR time sample at $k$-th position is characterized by the linear combination of $N$ exponential functions. Thus, in the FluoIR model, the free parameters are the scaling coefficients of the exponential functions, and their time constants. Nonetheless, according to the nature of the exponential functions, we adopt two strategies to model the $l$-th FluoIR time sample:

- **Local Approach**: the time constants of the exponential functions $\{\tau_{k,n}\}_{n=1}^{N}$ are different at each $k$-th spatial location:

$$\mathbf{h}_k[l] = c_{k,0} + \sum_{n=1}^{N} c_{k,n} e^{-\frac{l}{\tau_{k,n}}} \quad \forall l \in [0, L-1]. \tag{3}$$

- **Global Approach**: common exponential functions with time constants $\{\tau_n\}_{n=1}^{N}$ are scaled at each $k$-th spatial location:

$$\mathbf{h}_k[l] = c_{k,0} + \sum_{n=1}^{N} c_{k,n} e^{-\frac{l}{\tau_n}} \quad \forall l \in [0, L-1]. \tag{4}$$

The scaling coefficients $c_{k,n} \in \mathbb{R} \; \forall n \in [0, N]$ are selected different for each spatial location $k$, such that the estimated FluoD matches the actual measurement [19, 20, 29]. In this way, the *local perspective* in the FluoIR construction allows a major diversity in the fitting of the measured FluoDs, since at $k$-th spatial location there are $2N + 1$ parameters $\{c_{k,n}\}_{n=0}^{N}$ and $\{\tau_{k,n}\}_{n=1}^{N}$ to be estimated. On the other hand, the *global perspective* assumes the same exponential functions in the whole dataset, so at $k$-th spatial location just the $N + 1$ scaling parameters $\{c_{k,n}\}_{n=0}^{N}$ are obtained, resulting in a faster fitting procedure compared to the *local* approach at the expense of limited diversity. Nonetheless, the fitting accuracy of the measured FluoDs by the *local* and *global* approaches will depend on the studied FLIM dataset and the order selection of the multi-exponential models in Eqs (3) and (4).

The observation model in Eq (2) can be expressed in vector notation for the *local* approach as

$$\mathbf{y}_k = \mathbf{U} \mathbf{H}^{L}(\boldsymbol{\tau}_k) \mathbf{c}_k + \mathbf{v}_k \quad \forall k \in [0, K-1], \tag{5}$$

and similarly, for the *global* approach as

$$\mathbf{y}_k = \mathbf{U} \mathbf{H}^{G}(\boldsymbol{\tau}) \mathbf{c}_k + \mathbf{v}_k \quad \forall k \in [0, K-1], \tag{6}$$

where

$$\mathbf{U} = \begin{bmatrix} \mathbf{u}[0] & 0 & \dots & 0 \\ \mathbf{u}[1] & \mathbf{u}[0] & \dots & 0 \\ \vdots & \vdots & \ddots & \vdots \\ \mathbf{u}[L-1] & \mathbf{u}[L-2] & \dots & \mathbf{u}[0] \end{bmatrix} \in \mathbb{R}^{L \times L} \tag{7}$$

$$\mathbf{H}^L(\boldsymbol{\tau}_k) \quad = \begin{bmatrix} 1 & 1 & \ldots & 1 \\ 1 & e^{-\frac{1}{\tau_{k,1}}} & \ldots & e^{-\frac{1}{\tau_{k,N}}} \\ \vdots & \vdots & \ddots & \vdots \\ 1 & e^{\frac{L-1}{\tau_{k,1}}} & \ldots & e^{\frac{L-1}{\tau_{k,N}}} \end{bmatrix} \in \mathbb{R}^{L\times(N+1)} \tag{8}$$

$$\mathbf{c}_k \quad = [c_{k,0} \ldots c_{k,N}]^\top \in \mathbb{R}^{(N+1)} \tag{9}$$

$$\boldsymbol{\tau}_k \quad = [\boldsymbol{\tau}_{k,1} \ldots \boldsymbol{\tau}_{k,N}]^\top \in \mathbb{R}^N \tag{10}$$

$$\mathbf{v}_k \quad = \begin{bmatrix} \mathbf{v}_k[0] \ldots \mathbf{v}_k[L-1] \end{bmatrix}^\top \in \mathbb{R}^L \tag{11}$$

$$\mathbf{H}^G(\boldsymbol{\tau}) \quad = \begin{bmatrix} 1 & 1 & \ldots & 1 \\ 1 & e^{-\frac{1}{\tau_1}} & \ldots & e^{-\frac{1}{\tau_N}} \\ \vdots & \vdots & \ddots & \vdots \\ 1 & e^{\frac{L-1}{\tau_1}} & \ldots & e^{\frac{L-1}{\tau_N}} \end{bmatrix} \in \mathbb{R}^{L\times(N+1)} \tag{12}$$

$$\boldsymbol{\tau} \quad = [\boldsymbol{\tau}_1 \ldots \boldsymbol{\tau}_N]^\top \in \mathbb{R}^N \tag{13}$$

By following a similar perspective to [32], the InstR $\{\mathbf{u}[l]\}_{l=0}^{L-1}$ is assumed common to all $K$ spatial points in the FLIM dataset, and with a free-form. The InstR samples are also considered non-negative and normalized to sum-to-one to avoid scaling ambiguity, such that

$$\sum_{l=0}^{L-1}\mathbf{u}[l] = 1 \quad \& \quad \mathbf{u}[l] \geq 0 \quad \forall l \in [0, L-1]. \tag{14}$$

In FLIM applications, the InstR is sparse, so there is no need to estimate every sample. Hence, we consider only the first $\hat{L}$ terms ($\hat{L} < L$) to represent the InstR. As a result, the input matrix **U** in Eq (7) can be parametrized as a linear combination of $\hat{L}$ Toeplitz matrices:

$$\mathbf{U} = \sum_{l=0}^{\hat{L}-1}\theta_l\mathbf{U}_l^o, \tag{15}$$

where the parameter $\theta_l = \mathbf{u}[l]$ represents $l$-th sample in the InstR, and

$$\mathbf{U}_l^o = \mathbf{T}_{\mathbf{x}_l,\mathbf{z}_l} \in \mathbb{R}^{L\times L},$$

$$\mathbf{x}_l = \begin{bmatrix} \mathbf{0}_l & 1 & \mathbf{0}_{L-l-1} \end{bmatrix}^\top \in \mathbb{R}^L,$$

$$\mathbf{z}_l = \begin{bmatrix} \mathbf{x}_l[1] & \mathbf{0}_{L-1} \end{bmatrix}^\top \in \mathbb{R}^L.$$

With this mathematical description, the **blind deconvolution estimation** (BDE) problem is formulated as jointly obtaining the InstR components $\{\theta_l\}_{l=0}^{\hat{L}-1}$, the scaling coefficients $\{\mathbf{c}_k\}_{k=0}^{K-1}$

and the time constants, either $\{\tau_k\}_{k=0}^{K-1}$ or just $\tau$ for a given FLIM dataset $\mathcal{Y}$. Hence assuming Gaussian noise and independent measurements in Eq (5), we can formulate the *local* BDE as a maximum likelihood estimation, which is equivalent to the following nonlinear least-squares (NLS) approximation problem [30]:

$$\min_{\{\theta_l\}_{l=0}^{\hat{L}-1},\{\mathbf{c}_k\}_{k=0}^{K-1},\{\tau_k\}_{k=0}^{K-1}} \frac{1}{2}\sum_{k=0}^{K-1}\left\|\mathbf{y}_k - \sum_{l=0}^{\hat{L}-1}\theta_l \mathbf{U}_l^o \mathbf{H}^L(\tau_k)\mathbf{c}_k\right\|_2^2, \tag{16}$$

such that

$$\sum_l \theta_l = 1, \quad \& \quad \theta_l \geq 0 \quad \forall l \in [0, \hat{L}-1] \tag{17}$$

$$\mathbf{c}_k \geq 0, \quad \tau_k > 0 \quad \forall k \in [0, K-1]. \tag{18}$$

A similar approximation problem is defined for the *global* BDE:

$$\min_{\{\theta_l\}_{l=0}^{\hat{L}-1},\{\mathbf{c}_k\}_{k=0}^{K-1},\tau} \frac{1}{2}\sum_{k=0}^{K-1}\left\|\mathbf{y}_k - \sum_{l=0}^{\hat{L}-1}\theta_l \mathbf{U}_l^o \mathbf{H}^G(\tau)\mathbf{c}_k\right\|_2^2, \tag{19}$$

with restrictions in Eq (17), $\mathbf{c}_k \geq 0 \,\forall k$, and $\tau > 0$.

The inverse problems in Eqs (16) and (19) involve nonlinear interactions among the free parameters of the InstR $\{\theta_l\}_{l=0}^{\hat{L}-1}$, and those involved in the FluoIR $\{\mathbf{c}_k\}_{k=0}^{K-1}$, $\{\tau_k\}_{k=0}^{K-1}$ or $\tau$. To tackle these problems, we applied ALS approaches similar to [21] and [38, 39], where we estimate a solution for the FluoIR components while fixing the InstR, and vice versa until convergence.

## Alternated least-squares methodology for blind estimation

The proposed ALS methodology solves iteratively the local and global BDE problems in Eqs (16) and (19), and these two mathematical formulations have distinctive features that are discussed next. We observe that the InstR parameters $\{\theta_l\}_{l=0}^{\hat{L}-1}$ have a linear dependence on the approximation error, as well as the scaling coefficients $\{\mathbf{c}_k\}_{k=0}^{K-1}$ in the FluoIRs. However, the exponential time constants $\{\tau_k\}_{k=0}^{K-1}$ or $\tau$ show a nonlinear interaction. Furthermore, the scaling coefficients for each spatial point are always non-negative, while the time constants are restricted to be positive values. Moreover, in our formulation for FLIM datasets, the InstR is a narrow pulse without repetitions, so each measured FluoD will exhibit a sharp increase to its peak value, followed by a monotonic decrease. Moreover, all the measured FDs $\{\mathbf{y}_k[l]\}_{k=0}^{K-1}$ are scaled to sum-to-one, and also the InstR is limited to sum-to-one. As a result, the scaled shift symmetry described in [35] will not hold in our BDE formulation. In addition, at each iteration of the ALS scheme in the local and global approaches, a quadratic approximation problem is solved by either a non-linear least squares or linear least squares. So, at each iteration, the estimation error is reduced or at least maintained. Consequently, convergence is guaranteed in the iterative scheme, but only to a local minimum. For this reason, the initialization based on processing the FLIM dataset is a crucial step in our formulation to obtain meaningful results.

We set initial conditions for the InstR parameters, and for the time constants of the exponential functions by considering a raw estimation of the overall lifetime present in the dataset, this process is fully described in the following sections. To speed up the estimation of the InstR parameters, and considering that the InstR is common for all spatial points in the dataset, only

a random subset of FluoDs $\hat{\mathcal{Y}} \subset \mathcal{Y}$ ($\hat{K} = \text{card}(\hat{\mathcal{Y}}) < K$) was used for the initial estimation. The overall process for ALS assumes fixing one unknown parameter while optimizing for the other, i.e. fixing the InstR and estimating the FluoIRs or vice versa. In this sense, we first fixed the InstR to its initial condition and estimated the FluoIRs, for each spatial position of the reduced subset, by NLS following either the local or the global perspective [30]. Next, the estimated FluoIRs are fixed and the InstR is computed by constrained linear least squares. This alternated process is repeated until convergence for the InstR is reached. Finally, the FluoIRs are estimated taking into account the estimated InstR from the last iteration. The mathematical derivations for each optimization methodology are presented in the appendix, and the block diagrams of the local and global implementations are shown in Figs 2 and 3, respectively.

## Estimation of FluoIR parameters

At this stage, the components $\{\theta_l\}_{l=0}^{\hat{L}-1}$ of the InstR are assumed fixed, i.e. matrix $\mathbf{U}$ is known in Eqs (5) and (6). The *local* FluoIR estimation considers the estimation of the parameters $\{\mathbf{c}_k, \boldsymbol{\tau}_k\}$ at $k$-th spatial point by NLS with a Levenberg-Marquardt (LM) approach [30]. The damping term in the LM approach is adapted according to a similarity metric $\rho_k$ [40]. The mathematical

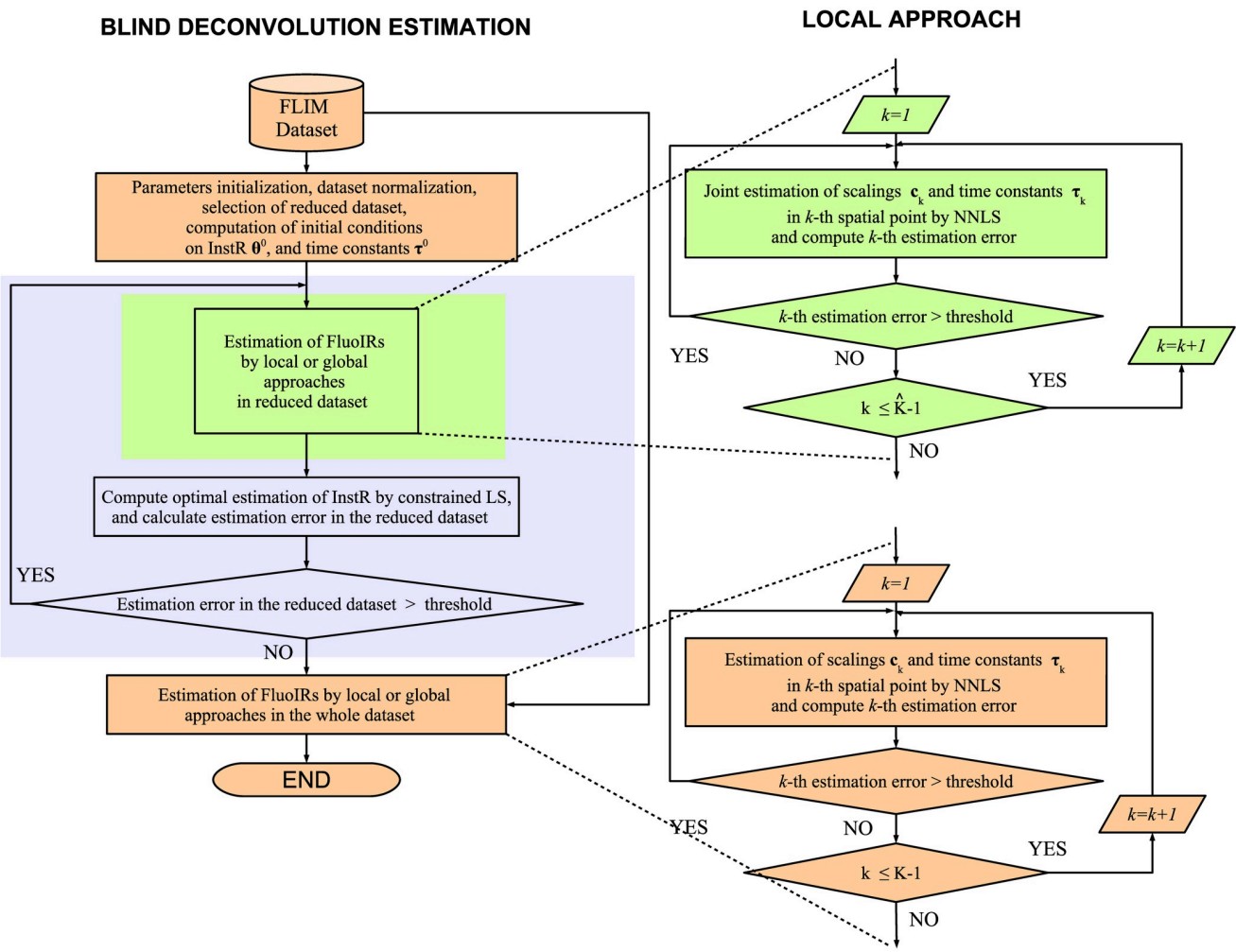

**Fig 2. Blind deconvolution estimation: Local approach.** Block diagram of the general methodology with local approach to estimate FluoIRs.

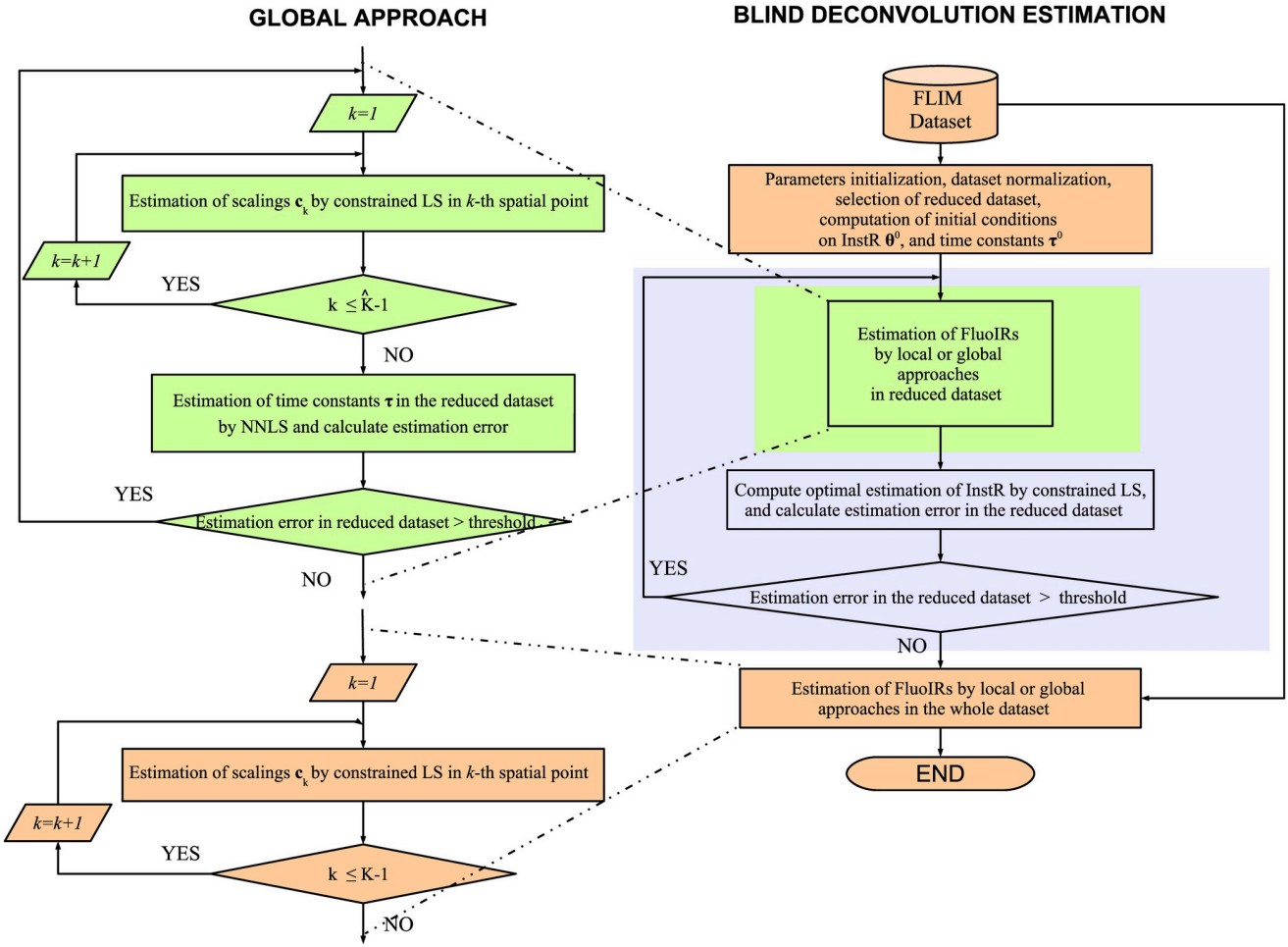

**Fig 3. Blind deconvolution estimation: Global approach.** Block diagram of the general methodology with global approach to estimate FluoIRs.

derivations of the local approach are described in S2 Appendix in S1 File (see Fig 2). For this purpose, we implement two iterative blocks for the $k$-th measurement to estimate the FluoIR parameters $(\mathbf{c}_k, \boldsymbol{\tau}_k)$:

1. A convergence loop that computes the ALS for the optimal parameters $(\mathbf{c}_k^h, \boldsymbol{\tau}_k^h)$ indexed by $h$ until the percentage approximation error $e_k^h$ converges with respect to the previous iteration $h - 1$:

$$e_k^h \triangleq \frac{\|\mathbf{y}_k - \check{\mathbf{y}}_k^h\|_2}{\|\mathbf{y}_k\|_2}, \tag{20}$$

where $\check{\mathbf{y}}_k^h = \mathbf{U}\,\mathbf{H}^L(\boldsymbol{\tau}_k^h)\,\mathbf{c}_k^h$ denotes the $k$-th estimated FluoD at $h$-iteration, i.e., the convergence criterion is given by

$$\frac{|e_k^{h+1} - e_k^h|}{e_k^h} < \epsilon_1, \tag{21}$$

where $\epsilon_1 > 0$ is the stopping threshold.

2. An internal loop that adjusts the damping factor $\lambda_k^h$ in the LM approach until $\rho_k^{h+1} > \epsilon_2$, where $\epsilon_2 > 0$. After this condition is achieved, the damping factor $\lambda_k^h$ is updated for subsequent iterations.

The *global* FluoIR estimation takes into consideration a non-negative least-squares (LS) approximation to estimate the scaling parameters $\{c_k\}$ for each $k$-th pixel in the dataset, which is followed by a NLS optimization for the time constants $\tau$ over the reduced dataset until convergence [30]. Similar to the local approach, the NLS optimization is implemented by a LM approach [40]. The mathematical derivations of the global approach are described in S3 Appendix in S1 File (see Fig 3). As previously outlined, we implement two iterative blocks to estimate the global FluoIR parameters:

1. An outer-loop that computes the ALS for the optimal scalings $\mathbf{C}^h = [\mathbf{c}_1^h \ldots \mathbf{c}_K^h]$ and time constants $\tau^h$ indexed both by $h$ until the percentage approximation error $e^h$ over the reduced dataset converge with respect to the previous iteration:

$$e^h = \frac{\|\mathbf{Y} - \mathbf{U}\mathbf{H}^G(\tau^h)\mathbf{C}^h\|_F}{\|\mathbf{Y}\|_F}, \tag{22}$$

where $\mathbf{Y} = [\mathbf{y}_1 \ \ldots \ \mathbf{y}\hat{K}] \in \mathbb{R}^{L \times \hat{K}}$ i.e.

$$\frac{|e^{h+1} - e^h|}{e^h} < \epsilon_1, \tag{23}$$

where $\epsilon_1 > 0$ is the stopping threshold.

2. An inner-loop that adjusts the damping factor $\lambda^h$ in the LM approach to estimate the time constants $\tau$ until $\rho^{h+1} > \epsilon_2$, where $\epsilon_2 > 0$. After this condition is achieved, the damping factor $\lambda^h$ is updated for subsequent iterations.

## Estimation of InstR parameters

At this step, the scaling coefficients $\{c_k\}$, and time constants $\{\tau_k\}$ or $\tau$ are assumed known and fixed in the cost functions in Eqs (16) or (19), and the optimization is implemented with respect to the samples of the InstR $\{\theta_l\}_{l=0}^{\hat{L}-1}$. In this case, a closed-form solution can be calculated, as will be shown next by using the reduced dataset $\hat{\mathcal{Y}}$. In this formulation, we are not assuming a pre-defined form or time-pattern for the InstR samples, just that $\{\theta_l\}_{l=0}^{\hat{L}-1}$ are always non-negative values and sum-to-one. The optimization problem is re-formulated in a compact structure as

$$\min_{\{\theta_l\}_{l=0}^{\hat{L}-1}} \underbrace{\frac{1}{2}\left\|\mathbf{Y} - \sum_{l=0}^{\hat{L}-1}\theta_l\mathbf{\Gamma}_l\right\|_F^2}_{\hat{J}}, \tag{24}$$

such that

$$\sum_l \theta_l = 1, \quad \& \quad \theta_l \geq 0 \ \ \forall l \in [0, \hat{L} - 1],$$

where

$$\mathbf{\Gamma}_l \triangleq \begin{cases} \mathbf{U}_{l^o[\mathbf{H}^L(\boldsymbol{\tau}_1)\mathbf{c}_1 \dots \mathbf{H}^L(\tau\hat{K})\mathbf{c}\hat{K}]} & \text{local approach} \\ \mathbf{U}_l^o \mathbf{H}^G(\boldsymbol{\tau})\mathbf{C} & \text{global approach} \end{cases} \tag{25}$$

i.e. $\mathbf{\Gamma}_l \in \mathbb{R}^{L \times \hat{K}}$ where $l \in [0, \hat{L} - 1]$. The mathematical derivations of the InstR parameter estimation are described in S4 Appendix in S1 File (see Figs 2 and 3). An iterative scheme following the ALS philosophy is implemented between the FluoIR estimation at each spatial pixel, and the InstR parameters over the reduced dataset. The iterative scheme is stopped after convergence of the overall parameters in the FluoIRs and InstR. If this iterative structure is indexed by $t$, i.e. $\{\theta_l^t, \mathbf{\Gamma}_l^t\}_{l=1}^{\hat{L}-1}$ the convergence performance is evaluated by a normalized metric:

$$\Upsilon^t \triangleq \frac{\|\mathbf{Y} - \sum_{l=0}^{\hat{L}-1} \theta_l^t \Gamma_l^t\|_F}{\|\mathbf{Y}\|_F}, \tag{26}$$

and the next stoppage criterion is considered as

$$\frac{|\Upsilon^{t+1} - \Upsilon^t|}{\Upsilon^t} < \epsilon_3, \tag{27}$$

where $\epsilon_3 > 0$ is the stopping threshold in this outer-loop.

## Initial conditions

An initial condition is required in the ALS scheme for InstR parameters $\theta^0$, and time constants in the FluoIRs estimation for the local $\{\tau_k^0\}_{k=0}^{\hat{K}-1}$ or global $\boldsymbol{\tau}^0$ approaches. We start by estimating an initial condition for $\theta^0$, which is defined as follows:

$$\theta^0[l] = \begin{cases} 0 & 0 \leq l \leq \dfrac{\hat{L}}{2} - 1 \\ 2\hat{L} & \dfrac{\hat{L}}{2} \leq l \leq \hat{L} - 1, \end{cases} \tag{28}$$

The proposed initial InstR represented by $\theta^0$ is a narrow pulse, like the one used in fluorescent measurements applications [32, 41], which will be refined in each subsequent iteration. Another advantage is that the resulting estimated InstR is aligned with the FluoDs, resulting in a better approximation of the FluoIRs.

An initial estimation for the time constants $\boldsymbol{\tau}^0$ or $\tau_k^0$ is computed from the average lifetime in the whole dataset, which is defined as:

$$\tilde{\tau}_k = \frac{\sum_{l=l_k^{max}}^{L-1} (l - l_k^{max})\mathbf{y}_k[l]}{\sum_{l=l_k^{max}}^{L-1} \mathbf{y}_k[l]} \quad \forall k \in [0, K-1], \tag{29}$$

where the peak value of each FluoD is defined as:

$$l_k^{max} = \arg \max_{l \in [0, L-1]} \mathbf{y}_k[l]. \tag{30}$$

By defining the mean and standard deviation of the values in $\{\tilde{\tau}_k\}_{k=0}^{K-1}$:

$$\bar{\tau} = (1/K)\sum_{k=0}^{K-1}\tilde{\tau}_k, \tag{31}$$

$$\sigma_\tau = (1/K)\sum_{k=0}^{K-1}(\tilde{\tau}_k - \bar{\tau})^2, \tag{32}$$

the elements in $\boldsymbol{\tau}^0$ and $\boldsymbol{\tau}_k^0$ are selected in the range $[\bar{\tau} - 3\sigma_\tau, \bar{\tau} + 3\sigma_\tau]$. By setting the initial condition on $\theta^0$, $\boldsymbol{\tau}_k^0$ for the local approach, the matrices $\mathbf{U}$ and $\mathbf{H}^L(\boldsymbol{\tau}^0)$ can be constructed from Eqs (7) and (8), respectively. To set the initial scaling coefficients $\mathbf{c}_k^0$ in the NLS scheme for $k$-th spatial pixel, a LS problem is formulated from Eq (16) as:

$$\min_{\mathbf{c}_k^0 \geq 0} \frac{1}{2}\|\mathbf{y}_k - \mathbf{U}\mathbf{H}^L(\boldsymbol{\tau}_k^0)\mathbf{c}_k^0\|_2^2, \tag{33}$$

which can be solved using standard numerical methods efficiently [30].

## Synthetic and experimental validation

To validate the proposed BDE algorithms, we consider synthetic and experimental FLIM datasets. In both cases, the performance of the proposals will be tested by measuring the estimation errors on the InstR and FluoDs; in addition, for the synthetic evaluation, we can also quantify the errors on the estimated FluoIRs. We also evaluate different scenarios of Gaussian and Poisson noise in the FluoDs for the synthetic datasets [23]. Furthermore, we quantify the shape of the InstR by the full-width at half-maximum (FWHM) parameter $\Delta\mathbf{u}_{fwhm}$:

$$\Delta\mathbf{u}_{fwhm} = l_2 - l_1, \tag{34}$$

where the time indexes $0 < l_1 < I_{max}$ and $I_{max} < l_2 < L - 1$ satisfy

$$\mathbf{u}[l_1] = \mathbf{u}[l_2] = u_{max}/2, \tag{35}$$

where

$$I_{max} = \arg\max_{l\in[0,L-1]}\mathbf{u}[l] \quad \& \quad u_{max} = \mathbf{u}[I_{max}]. \tag{36}$$

By assuming that $\check{\mathbf{y}}_k$, $\check{\mathbf{h}}_k$ and $\check{\mathbf{u}}$ denote the FluoD, FluoIR and InstR estimations by a BDE algorithm, respectively, and $\mathbf{y}_k$, $\mathbf{h}_k$ and $\mathbf{u}$ the corresponding ground-truths, we employ four estimation performance metrics:

$$E_y = \frac{1}{K}\sum_{k=0}^{K-1}\frac{\|\mathbf{y}_k - \check{\mathbf{y}}_k\|_2}{\|\mathbf{y}_k\|_2},$$

$$E_h = \frac{1}{K}\sum_{k=0}^{K-1}\frac{\|\mathbf{h}_k - \check{\mathbf{h}}_k\|_2}{\|\mathbf{h}_k\|_2}, \tag{37}$$

$$E_u \quad = \frac{\|\mathbf{u} - \check{\mathbf{u}}\|_2}{\|\mathbf{u}\|_2},$$

$$E_{fwhm} \quad = \frac{|\Delta\mathbf{u}_{fwhm} - \Delta\check{\mathbf{u}}_{fwhm}|}{\Delta\mathbf{u}_{fwhm}}. \tag{38}$$

The metrics $(E_u, E_h, E_y)$ provide information of the estimation performance with respect to the estimated InstR, FluoIRs and FluoDs in a percentage fashion. For the synthetic FluoIR $\mathbf{h}_k$ at $k$-th spatial point, the average lifetime (ALT) $\boldsymbol{\tau}_k$ is a key parameter related to the fluorescent property of the sample:

$$\boldsymbol{\tau}_k = \frac{\mathbf{t}^\top \mathbf{h}_k}{\mathbf{1}_L^\top \mathbf{h}_k} \qquad k \in [0, K-1], \tag{39}$$

where $\mathbf{t} = [0\ T\ \dots\ (L-1)T]^\top \in \mathbb{R}^L$ represents a vector of the sampling times. These ALTs are used to generate a representative image with spatial information of the chemical composition of the sample. Hence, the error between the ALTs generated through the synthetic FluoIR $\{\tau_k\}_{k=0}^{K-1}$ and the estimated one $\{\check{\tau}_k\}_{k=0}^{K-1}$ can be quantified by using the next normalized metric:

$$E_{alt} = \frac{1}{K} \sum_{k=0}^{K-1} \frac{|\tau_k - \check{\tau}_k|}{\tau_k}. \tag{40}$$

In our evaluations, we compare the proposed BDE local and global algorithms based on multi-exponential models (BDELME and BDEGME) to the BDE by a Laguerre-basis (BDELB) [32], and a blind extension of the exponentials library deconvolution (BDEEL) approach [42]. In addition, we also compare the standard local and global deconvolution methodologies with a multi-exponential model that assume available the InstR, and they are denoted as: DELME and DEGME, respectively [17, 18]. As suggested in [32], BDELB was implemented with a 8th order approximation and shape parameter 0.85. Meanwhile, for BDEEL, the exponential library contains 25 elements and a weight factor of 0.25, as suggested in [41] and [42]. All the MATLAB implementations of the methodologies: DELME, DEGME, BDELME, BDEGME, BDELB and BDEEL are freely available in the website http://galia.fc.uaslp.mx/~bde.

## Synthetic evaluation

The proposals were first validated by using synthetic datasets under different types and noise levels. The synthetic datasets were generated considering a measured InstR [5], with a sampling interval $T = 0.25$ ns and a length of 186 samples ($L = 186$). This measured InstR $\{\mathbf{u}[l]\}_{l=0}^{L-1}$ is a positive time-signal with a sharp rising time and exponential decay with $\Delta\mathbf{u}_{fwhm} = 1.53$ ns. The $k$-th synthetic FluoIR is modeled as a sum of $N_{synth} \in \{2, 3, 4\}$ exponential functions:

$$\mathbf{h}_k[l] = a_{k,0} + \sum_{i=1}^{N_{synth}} a_{k,i} e^{-l\frac{T}{\tau_{k,i}}} \quad \forall k \in [0, K-1],\ l \in [0, L-1], \tag{41}$$

where the magnitudes $a_{k,i}$ were selected to have uniform regions of high concentration of all components in the dataset [41], $a_{k,0} = 0.001\ \forall k$, and the characteristic times are selected randomly, but in a limited interval $\tau_{k,1} \sim \mathcal{U}[3.75, 4.25]$ ns, $\tau_{k,2} \sim \mathcal{U}[8.75, 9.25]$ ns, $\tau_{k,3} \sim \mathcal{U}[1.25, 1.75]$ ns, and $\tau_{k,4} \sim \mathcal{U}[6.25, 6.75]$ ns $\forall k$. With these definitions of the synthetic FluoIRs, the BDE local and global perspectives are both viable.

Next, the synthetic noise-free FluoDs $\mathbf{y}_k^o[l]$ are obtained by applying the convolution operator in (2), i.e. $\mathbf{y}_k^o[l] = \mathbf{u}[l] \star \mathbf{h}_k[l]$. In our evaluation, we included Gaussian and Poisson noise to the FluoDs to take into account uncertainty in the equipment according to the following model [23]:

$$\mathbf{y}_k[l] = \mathbf{y}_k^o[l] + \omega_k[l] + \sqrt{\mathbf{y}_k^o[l]} \cdot \epsilon_k[l] \quad \forall l \in [0, L-1], \tag{42}$$

where $\omega_k[l] \sim \mathcal{N}(0, \sigma_{G,k}^2)$ and $\epsilon_k[l] \sim \mathcal{N}(0, \sigma_{P,k}^2)$ represent normal random variables, and the variances $\sigma_{G,k}^2$, $\sigma_{P,k}^2$ are selected with respect to a desired signal-to-noise ratio (SNR) and peak-to-noise signal ratio (PSNR):

$$\text{SNR} = 10 \log_{10} \frac{\|\mathbf{y}_k^o\|^2}{\sigma_{G,k}^2} \quad \forall k \in [0, K-1], \tag{43}$$

$$\text{PSNR} = 10 \log_{10} \frac{\max_{l \in [0,L-1]} (\mathbf{y}_k^o[l])^2}{\sigma_{P,k}^2} \quad \forall k \in [0, K-1]. \tag{44}$$

Since the construction of the synthetic datasets involved uniformly distributed random samples, we carried out a Monte Carlo evaluation with 10 repetitions by implementing the BDE algorithms according to the parameters listed in Table 1, whose selection is explained next. In the synthetic evaluation, a spatial domain of $100 \times 100 = 10,000$ samples were generated by Eqs (41) and (42) at different values of SNR $\in \{40, 45, 50, 55\}$ dB, and PSNR $\in \{10, 15, 20, 25\}$ dB, and the resulting datasets were analyzed by the BDE algorithms, i.e. $K = 10,000$. The random spatial sampling retained only $\hat{K} = 2,000$ to estimate the InstR, i.e. 20% of the complete dataset. As previously mentioned, we considered three different values for the number of exponential functions in the synthetic FluoIRs $N_{synth} \in \{2, 3, 4\}$. Since for DELME, DEGME, BDELME and BDEGME, we could not know a priori the number of exponentials in the FluoIRs model, we assume a fourth order model in the synthetic datasets, i.e. $N = 4$.

First, we have included a numerical convergence evaluation of the BDE schemes (BDELME and BDEGME) with the synthetic datasets for different noise levels, and orders in the multi-exponential model of the synthetic impulse response. The stopping threshold is set to $\epsilon_3 = 1 \times 10^{-3}$. The resulting normalized metric $Y^t$ in the alternated least squares iterations is described below in Fig 4, where it is observed that in either BDE scheme, the convergence is always monotonic for any noise combination and model order, and just its final steady-state value depends on the noise level. From Fig 4, we conclude that the local approach (BDELME) converges slower than the global scheme (BDEGME), which could be intuitively expected,

**Table 1. Parameters of synthetic dataset and BDE implementation during the synthetic evaluation.**

| Parameter | Value |
|---|---|
| $T$ | 0.25 ns |
| $K$ | 10,000 |
| $L$ | 186 |
| $N_{synth}$ | {2,3,4} |
| $\tau_{min}$ | 0.5 ns |
| $\tau_{max}$ | 15.0 ns |
| $\epsilon_1$ | 0.05 |
| $\epsilon_2$ | 0.05 |
| $\epsilon_3$ | 0.05 |

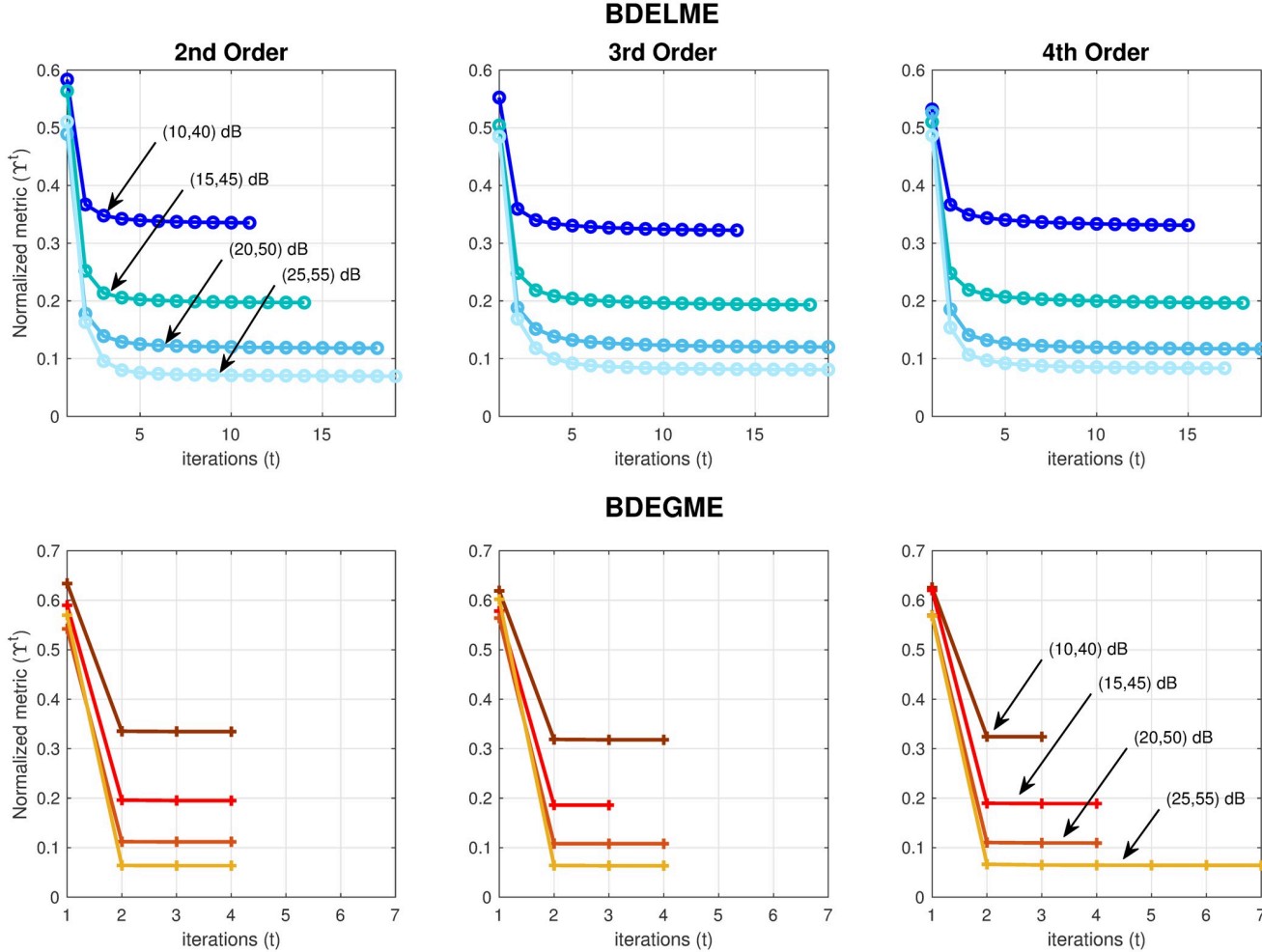

**Fig 4. Normalized metric ($\Upsilon^t$) vs iteration ($t$) in the ALS scheme for (top row) BDELME and (bottom row) BDEGME by considering synthetic FLIM datasets at different noise levels (SNR,PSNR) $\in$ {(10, 40), (15, 45), (20, 50), (25, 55)} dB, and impulse response orders (2nd, 3rd and 4th).**

since BDELME has roughly twice the free parameters to optimize. For BDELME, the most drastic reductions are achieved in the first five iterations, and for BDEGME, in the first two.

In the following evaluations, to evaluate the convergence in the iterative process of the BDE algorithms (see Figs 2 and 3), we use a 5% tolerance in all iterations, $\epsilon_1 = \epsilon_2 = \epsilon_3 = 0.05$, as a good balance between precision and complexity in our evaluations. All the data processing was carried out in MATLAB, and the computational time was measured using the function "*tic-toc*". For the experiments, we used a MacBook Pro with an Intel Dual Core i5 CPU at 2.3 GHz, and 16 GB of RAM. In addition, our evaluation also considered DELME and DEGME with the measured InstR $\{\mathbf{u}[l]\}_{l=0}^{L-1}$ by applying the procedures in S1 and S3 Appendices in S1 File. Hence, the performance indexes $E_y$, $E_h$ and $E_{alt}$ can examine the estimation accuracy in the FluoDs, FluoIRs and ALTs by a direct deconvolution process, as well as the computational time. Meanwhile, the metrics $E_u$ and $E_{fwhm}$ can also quantify the accuracy in the InstR estimations by the BDE strategies.

Fig 5 illustrates the computational time and performance metrics ($E_y$, $E_h$, $E_{alt}$) as a function of ascending SNR/PSNR pairs for 2nd, 3rd and 4th order synthetic FluoIR models. As expected, the lowest computational time is achieved by the standard deconvolution techniques,

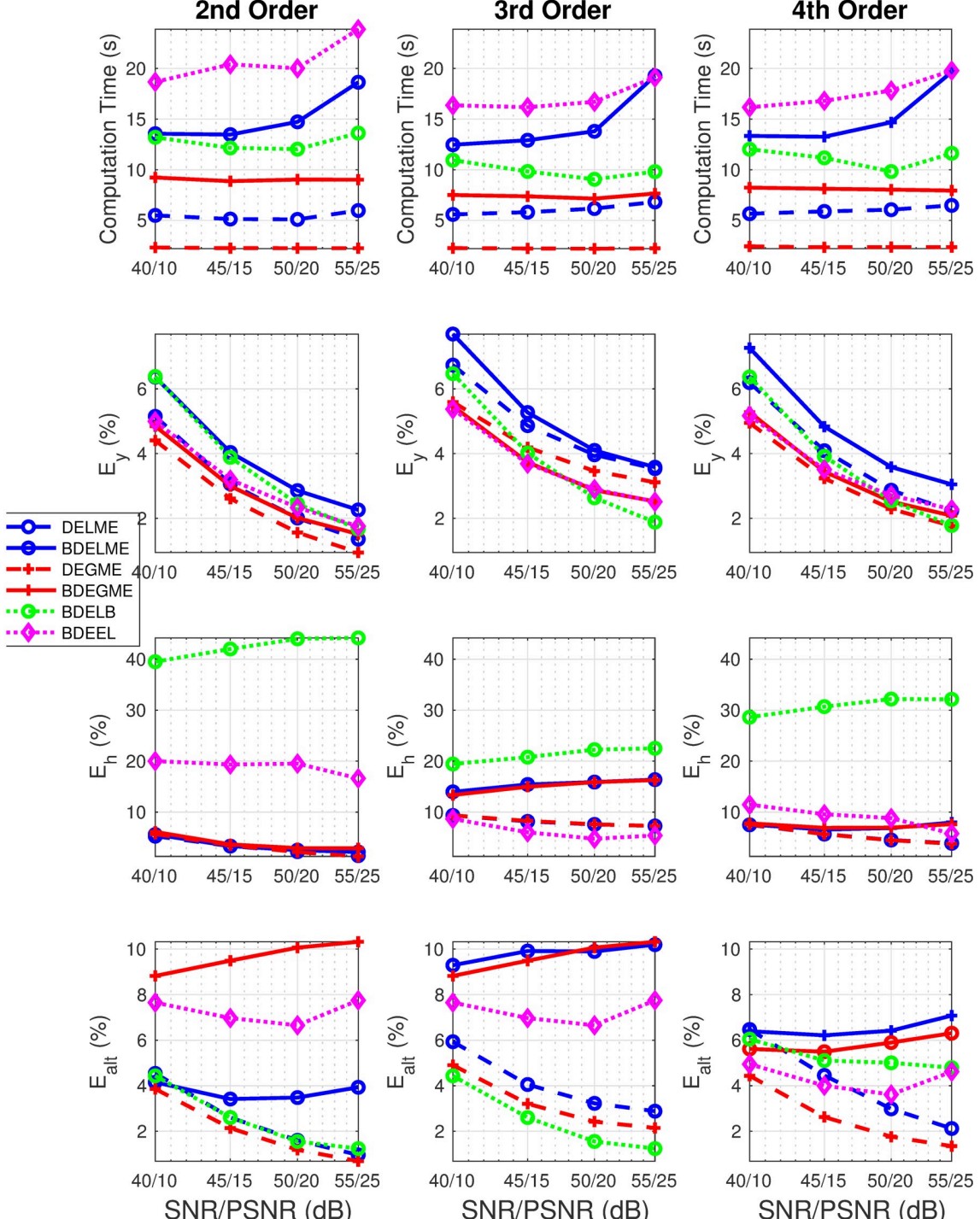

**Fig 5. Performance metrics in synthetic evaluation with 2nd, 3rd and 4th order FluoIR models: (top row) Computational time, (middle-top row) $E_y$, (middle-bottom row) $E_h$, and (bottom row) $E_{alt}$.**

DELME and DEGME. In fact, the results in the computational time are independent of the order in the synthetic model. Now, from the BDE strategies, BDEGME obtained the lowest computational time consistently, and BDELME was just surpassed by BDEEL in all scenarios. For the $E_y$ metric, the results had the same tendency for all algorithms, as the pair SNR/PSNR

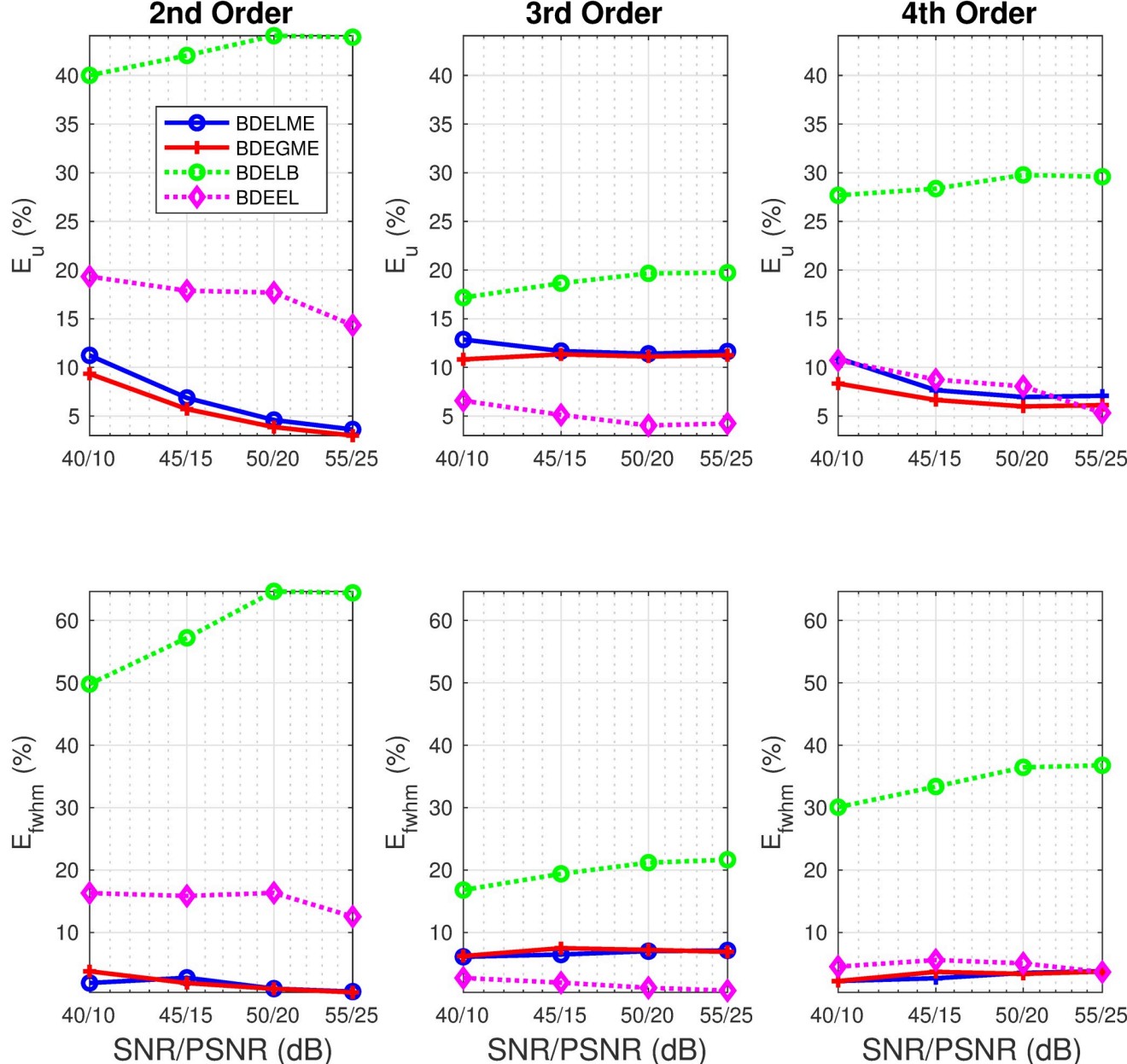

**Fig 6. Performance metrics in synthetic evaluation with 2nd, 3rd and 4th order FluoIR models: (top-row) $E_u$, and (bottom row) $E_{fwhm}$.**

increased, the error metric decreased. In the FluoIR estimations, BDELB reached the worst performance in all scenarios. For all BDE algorithms, the results for $E_h$ show a small variability with respect to the SNR/PSNR pairs. The proposed BDELME and BDEGME had always $E_h$ errors lower than 20%, in contrast with state of the art approaches. Finally, the errors in ALT $E_{alt}$ are small (<10%) for all methodologies, despite all different noise types and levels. Fig 6 shows the estimation performance in the InstR by the metrics $E_u$ and $E_{fwhm}$. It is remarkable that the proposed blind techniques BDELME and BDEGME reached the best performance for all noise pairs SNR/PNSR in 2nd and 4th order synthetic models. Meanwhile, for the 3rd

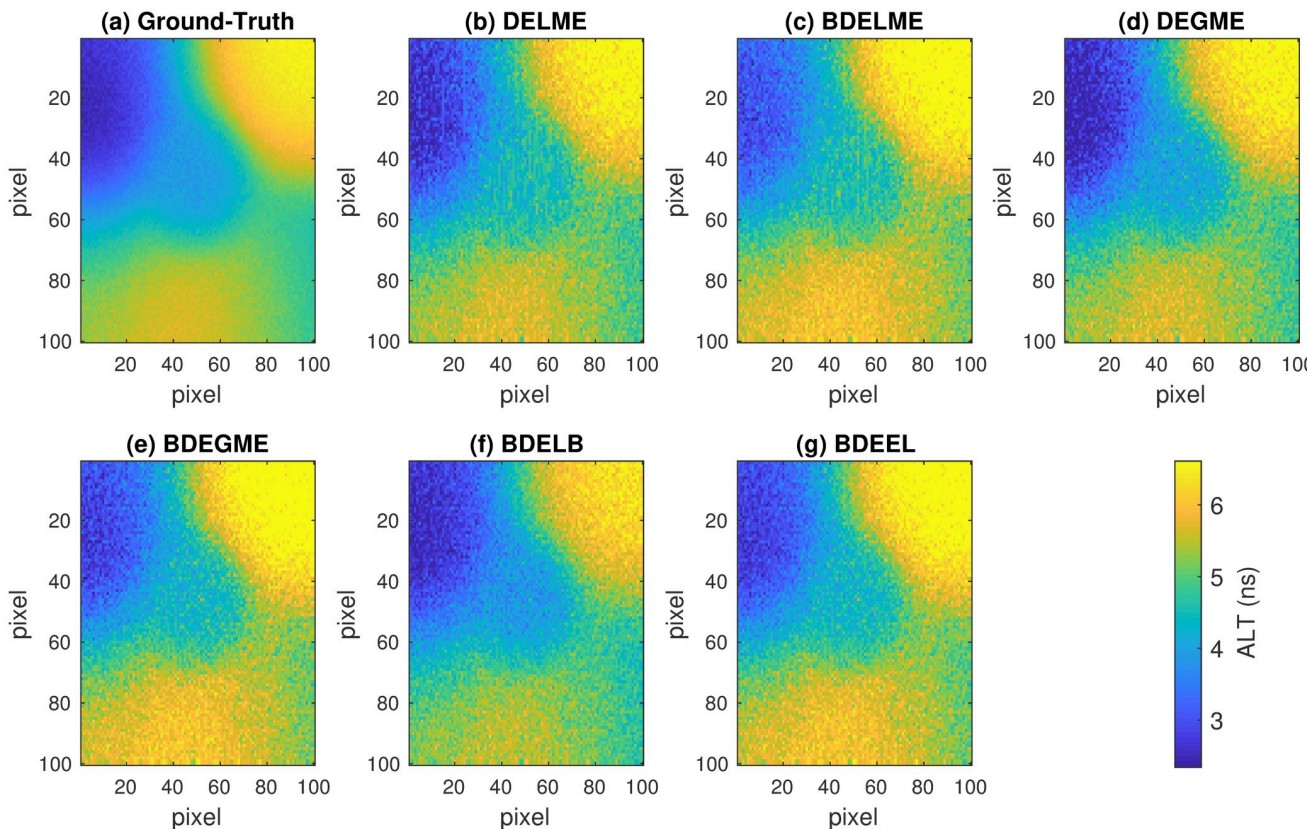

**Fig 7. Average lifetime map for one realization in synthetic evaluation with 3rd order FluoIR model, SNR = 40 dB and PSNR = 10 dB: (a) Ground-truth, (b) DELME, (c) BDELME, (d) DEGME, (e) BDEGME, (f) BDELB, and (g) BDEEL.**

order model, their performance was in between the worst by BDELB and the minimum by BDEEL.

To illustrate the estimation performance, Fig 7 shows the ALT map for one realization of the synthetic dataset with the most severe noise conditions, i.e. SNR = 40 dB and PSNR = 10 dB, and 3rd order synthetic model. As shown in Fig 6, all the ALT maps are consistent with the ground-truth, since the errors are less than 10% for all deconvolution techniques. Finally, Fig 8 presents the resulting time responses for $k = 1,000$ spatial points in the same testing scenario. Thus, the top plot highlights the heavy noise condition in the synthetic FluoD measurement, and the accurate estimation by all techniques, since the errors are lower than 7%, as shown in the middle column for $E_y$. The middle plot illustrates that the closest response to the ground-truth is achieved by DELME, BDELME, DEGME and BDEGME. Finally, the bottom plot shows that the InstR is accurately estimated by all BDE methodologies, since the errors are always lower than 20% (see Fig 6).

As conclusions of the synthetic evaluation, BDEGME reached the fastest convergence in the blind techniques, independently to the noise scenario. BDELME required more computational time, but not significantly more to BDEGME. In the FluoIR and InstR estimations, BDEGME and BDELME reached the best performance in the 2nd and 4th synthetic order models, and for the 3rd order model, their results were between the best and worst of all the approaches considered.

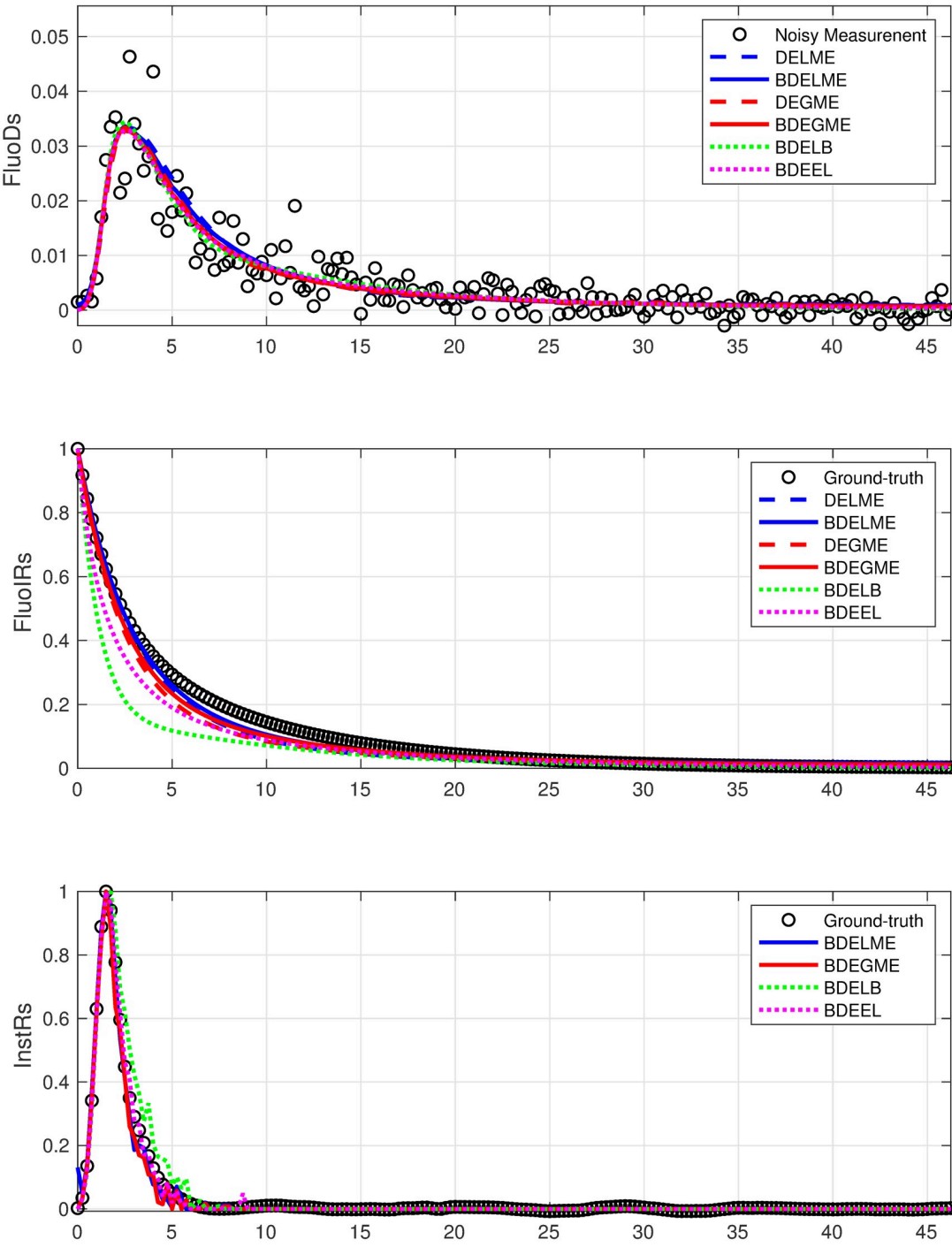

**Fig 8. Estimated time responses for $k = 1,000$ spatial position for one realization with 3rd order FluoIR model, SNR = 40 dB and PSNR = 10 dB: (top) FluoDs, (middle) FluoIRs, and (bottom) InstRs.**

## Experimental evaluation with fluorescence dyes

The proposals were first validated experimentally with fluorescence dyes: POPOP, FAD and NADH. The lifetimes of these dyes are reported in the literature: $\sim 1.3$ ns, 2.0-2.5 ns and 0.3-

**Table 2. Estimated average lifetimes (ns) for synthetic dyes.**

| Synthetic Dye | BDELME | BDEGME | BDELB | BDEEL | Literature |
|---|---|---|---|---|---|
| POPOP (390±20 nm) | 1.409 ± 0.076 | 1.685 ± 0.061 | 1.678 ± 0.053 | 1.467± 0.057 | 1.3 ns |
| NADH (390±20 nm) | 0.558 ± 0.063 | 0.749 ± 0.075 | 0.888 ± 0.042 | 0.617 ± 0.050 | [0.3,0.6] ns |
| POPOP (494±41 nm) | 1.039 ± 0.121 | 1.137 ± 0.070 | 1.216 ± 0.033 | 1.046 ± 0.028 | 1.3 ns |
| FAD (494±41 nm) | 2.976 ± 0.073 | 2.976 ± 0.073 | 3.168 ± 0.074 | 3.084 ± 0.074 | [2.4,2.9] ns |

0.6 ns, respectively [1, 4]. The FLIM datasets were collected at the wavelength channels: 390 ±40 nm (channel 1), and 494±41 nm (channel 2), with a sampling time of 0.25 ns ($T = 0.25$ ns). The FAD dye has an emitting response only in channel 1, the POPOP only in channel 2, while the NADH response to both channels. For each channel measurement, there are defined 80 time samples ($L = 80$) over a spatial resolution of 1,000 pixels ($K = 1,000$). Since the number of spatial samples is relatively small, there is no random subsampling to estimate the InstR, i.e. $\hat{K} = K$. The InstR was measured at both channels, where the FWHM of the UV laser-pulse (355 nm) was $\Delta \mathbf{u}_{fwhm} = 1.78$ ns and 1.97 ns for channel 1 and 2, respectively. Since we expect two fluorophores per channel, we set $N = 2$ for BDELME and BDEGME. The estimated ALT by using BDELME, BDEGME, BDELB and BDEEL are presented in Table 2 with parameters $\tau_{min} = 0.25$ ns and $\tau_{max} = 4$ ns (based on the literature results [1, 4]). The computational times for the BDE algorithms in channels 1/channel 2 were 1.78/2.76 s, 1.75/2.07 s, 2.95/2.80 s, and 2.45/2.92 s for BDELME, BDEGME, BDELB and BDEEL, respectively. Thus, the lowest computational times were obtained by the proposed algorithms: BDELME and BDEGME.

These results show a good agreement with the literature by considering that there is an uncertainty factor in the estimation due to the sampling interval of 0.25 ns. In addition, the performance of the proposed BDE algorithms is visualized by using the Bland-Altman (B&A) methodology with the estimated ALTs [43]. The B&A plot measures the similarity between two types of data, by setting limits of agreement in terms of the mean and the standard deviation (SD) of the differences between the two sources. In this case, the B&A analysis was computed with the ALTs generated by pairs of the studied BDE algorithms: BDELME, BDEGME, BDELB and BDEEL, and Figs 9 and 10 illustrate the B&A plots for all pairs. In addition at the top of each subplot in Figs 9 and 10, the correlation coefficient $\rho$ was computed between both estimated ALTs. As a result, we observe that in all scenarios, two distinctive ALTs are obtained per channel by the two characteristic clusters in the B&A plots. Also, the plots illustrate that the estimated ALTs are mostly contained in the 95% confidence interval defined by the red dashed-lines (mean ± 1.96 SD), and the correlation coefficients are always greater than 0.99 for both channels.

## Experimental evaluation with oral tissue samples

From the study in [44], oral tissue samples were used for validation in this section that belong to different regions in the oral cavity. Dysplastic and cancerous oral lesions were analyzed by *in vivo* clinical endogenous mFLIM images. The imaging protocol was approved by the Institutional Review Board at Texas A&M University. The clinical diagnosis and more detailed information about the samples are presented in Table 3. The temporal resolution of the measurements is 0.16 ns ($T = 0.16$ ns). All the measurements included the fluorescent responses to three wavelength bands: 390 ± 20, 452 ± 22.5, and 550 ± 20 nm, that correspond to channel 1, channel 2 and channel 3, respectively. Only 186 time samples were considered for each channel ($L = 186$). The spatial dimensions of the tissues are approximately 10 mm × 10 mm, divided equiespatially in 160 × 160 pixels ($K = 25,600$). Due to low SNR, some pixels in the images

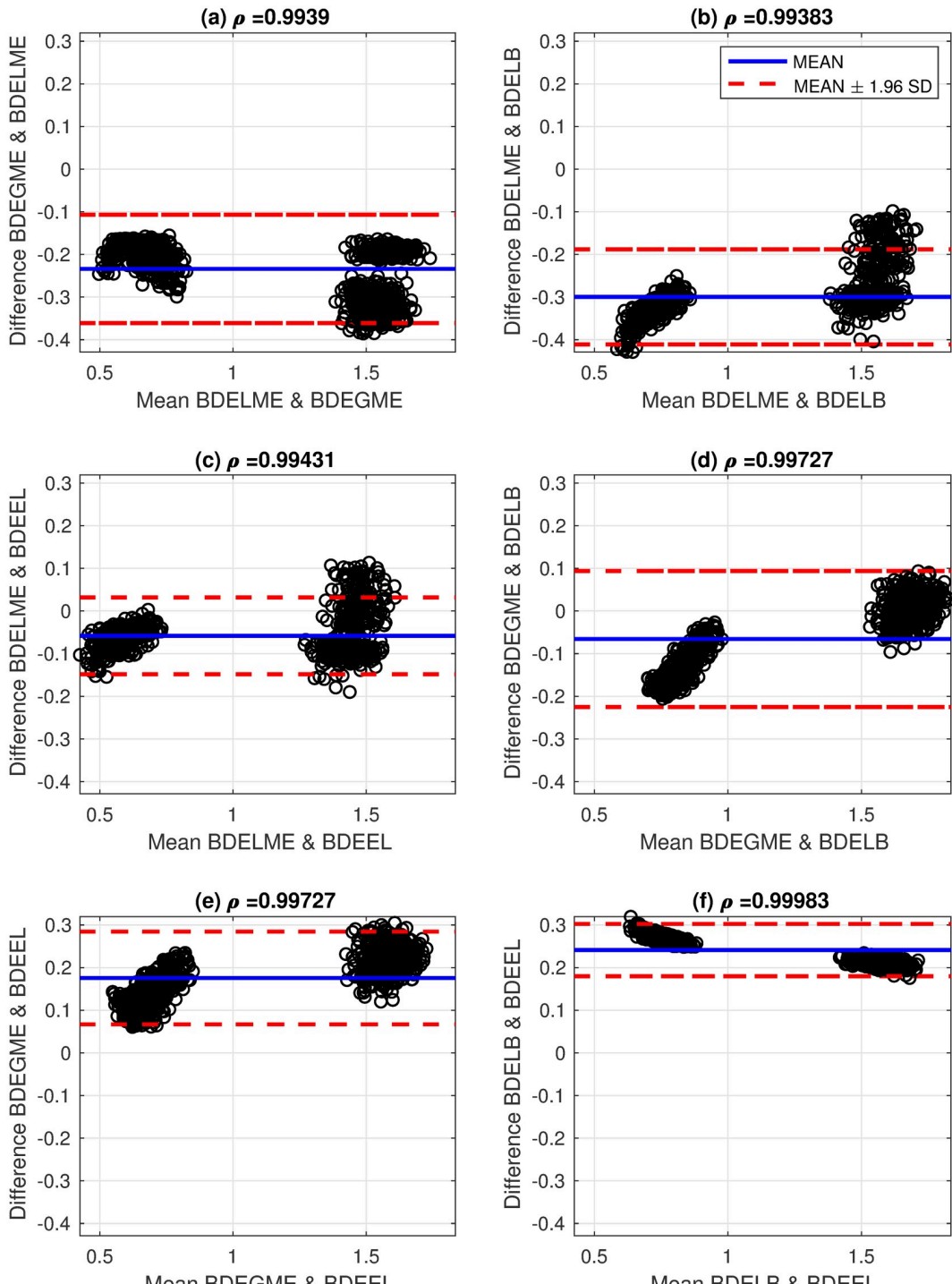

**Fig 9. Bland-Altman plots for estimated average lifetimes with fluorescence dyes in channel 1 (390±20 nm) and correlation coefficients: (a) BDELME vs BDEGME, (b) BDELME vs BDELB, (c) BDELME vs BDEEL, (d) BDEGME vs BDELB, (e) BDEGME vs BDEEL, and (f) BDELB vs BDEEL.**

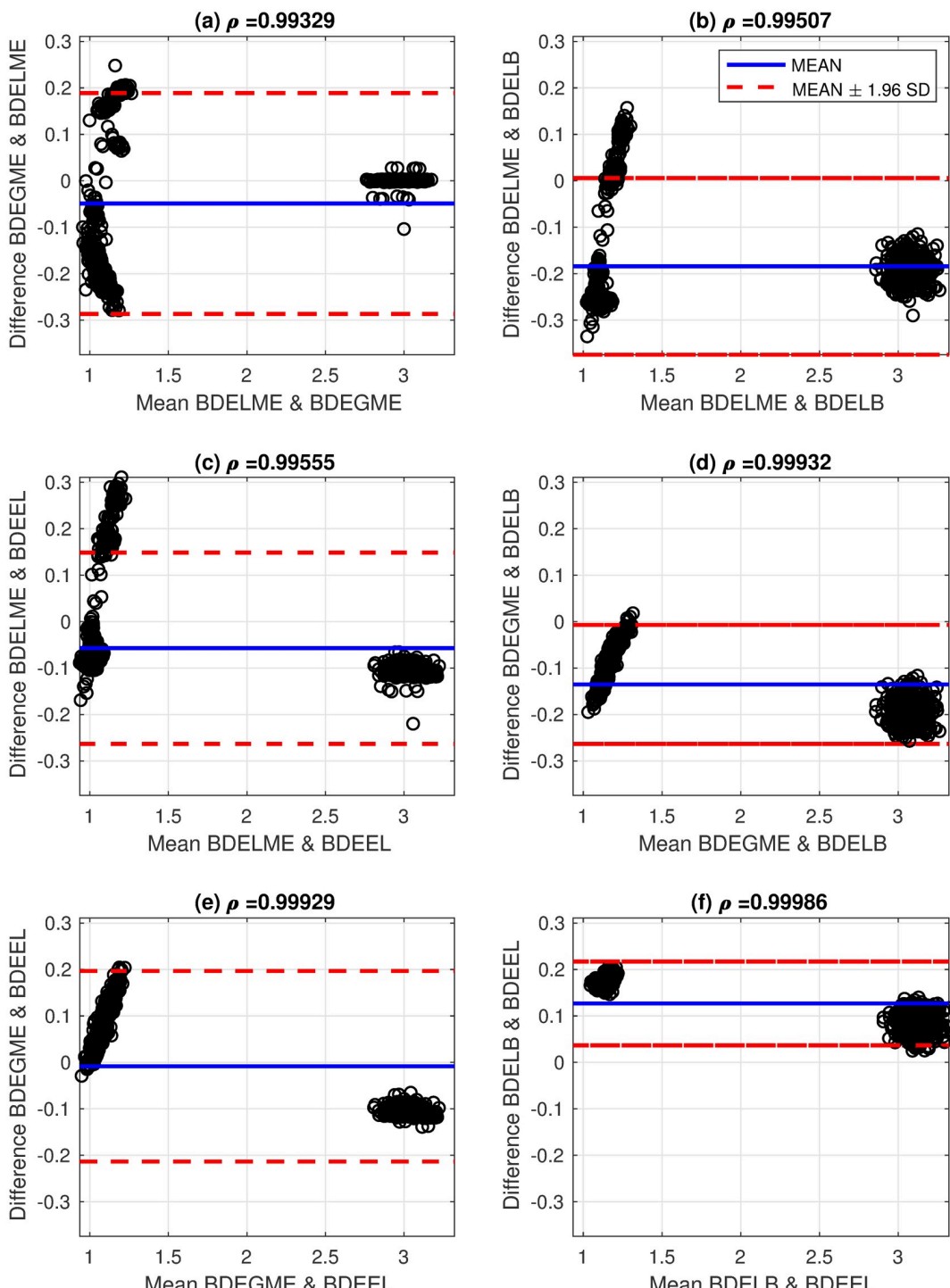

**Fig 10. Bland-Altman plots for estimated average lifetimes with fluorescence dyes in channel 2 (494±41 nm) and correlation coefficients:** (a) BDELME vs BDEGME, (b) BDELME vs BDELB, (c) BDELME vs BDEEL, (d) BDEGME vs BDELB, (e) BDEGME vs BDEEL, and (f) BDELB vs BDEEL.

**Table 3. Detailed information of the oral tissue lesion samples.**

| Sample | Number of Pixels | Region | Medical diagnosis |
|--------|------------------|--------|-------------------|
| 49 | 17,693 | Gingiva | Squamous cell carcinoma |
| 62 | 23,918 | Gingiva | Squamous cell carcinoma |
| 65 | 25,429 | Tongue | Dysplasia |
| 69 | 23,961 | Gingiva | Benign |
| 82 | 25,173 | Gingiva | Benign |

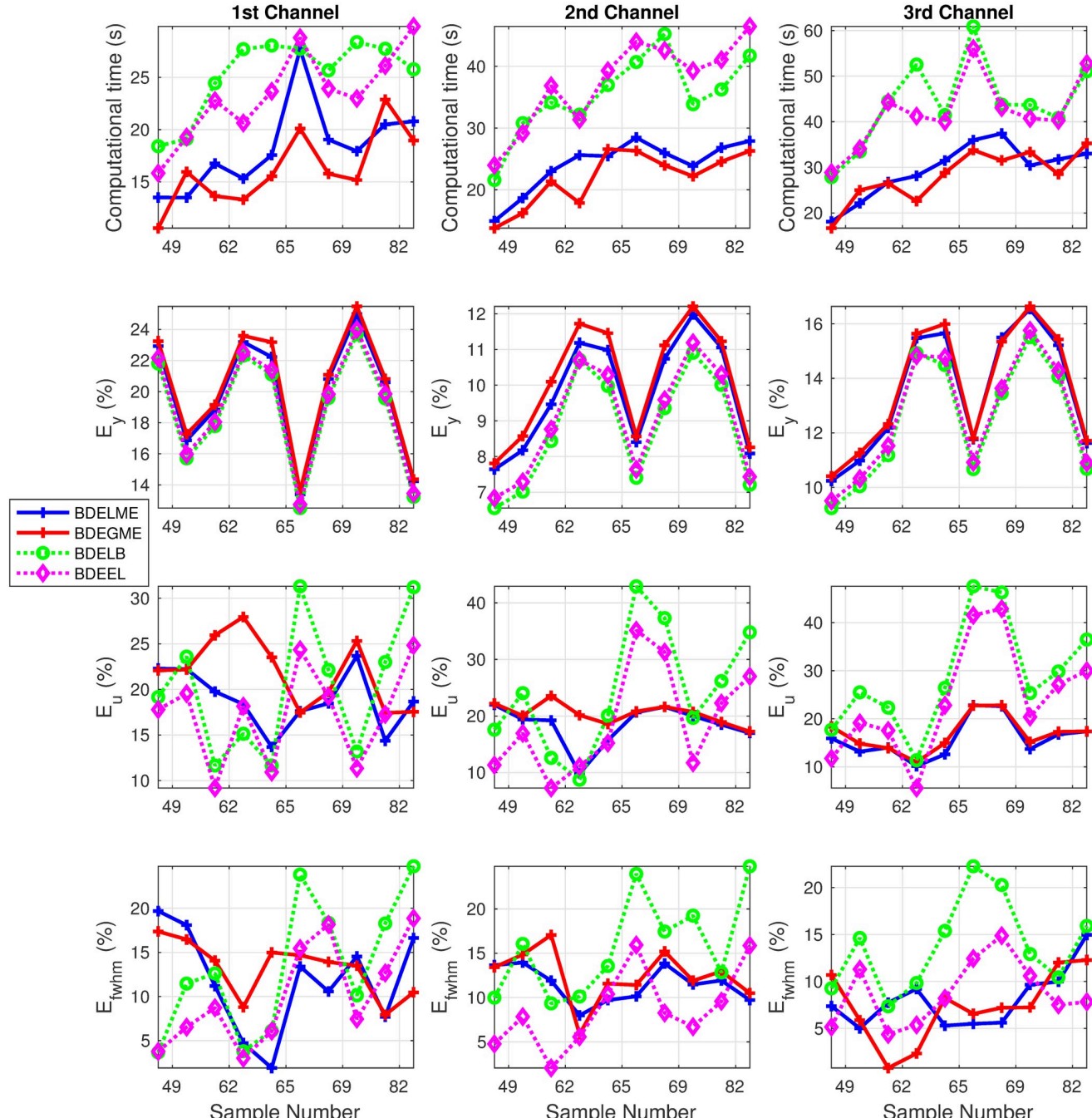

**Fig 11. Performance metrics in experimental evaluation with lesion and normal samples for the three spectral channels: (top row) Computational time, (middle-top row) $E_y$, (middle-bottom row) $E_u$, and (bottom row) $E_{fwhm}$.**

were manually masked, so their information will not misguide the BDE strategies. During the experimental evaluations, the following parameters were set for BDELME and BDEGME as $\tau_{min} = 0.64$ ns, $\tau_{max} = 16$ ns, and $\epsilon_1 = \epsilon_2 = \epsilon_3 = 0.05$. The experimental InstRs used to generate the fluorescent measurements in channels 1, 2 and 3 have a FWHM of $\Delta\mathbf{u}_{fwhm} = 1.70$ ns, 1.88 ns and 2.21 ns, respectively. For each oral tissue the dataset contains two samples, lesion and reference. The reference sample was taken from the symmetrical location of the sagittal plane.

Fig 11 shows the resulting computational time and performance metrics ($E_y$, $E_u$, $E_{fwhm}$) for all the datasets describe in Table 3. We can observe that the computational time is always the lowest for BDELME and BDEGME for all three channels. With respect to the estimation error in the FluoDs, $E_y$ exhibits the same tendency for all datasets and channels, where the differences are lower than 1%. Meanwhile, for the InstR estimations, no particular BDE methodology achieved consistently the lowest errors $E_u$ and $E_{fwhm}$. Nonetheless, BDELME and BDEGME presented more regular responses in $E_u$ and $E_{fwhm}$, especially in the second and third channels. In general, the largest errors were obtained by BDELB in the three channels.

To illustrate a specific response for the estimated FluoIRs, Fig 12 presents the estimated ALT maps for lesion sample No. 82 in the 3rd channel with the four BDE algorithms. The

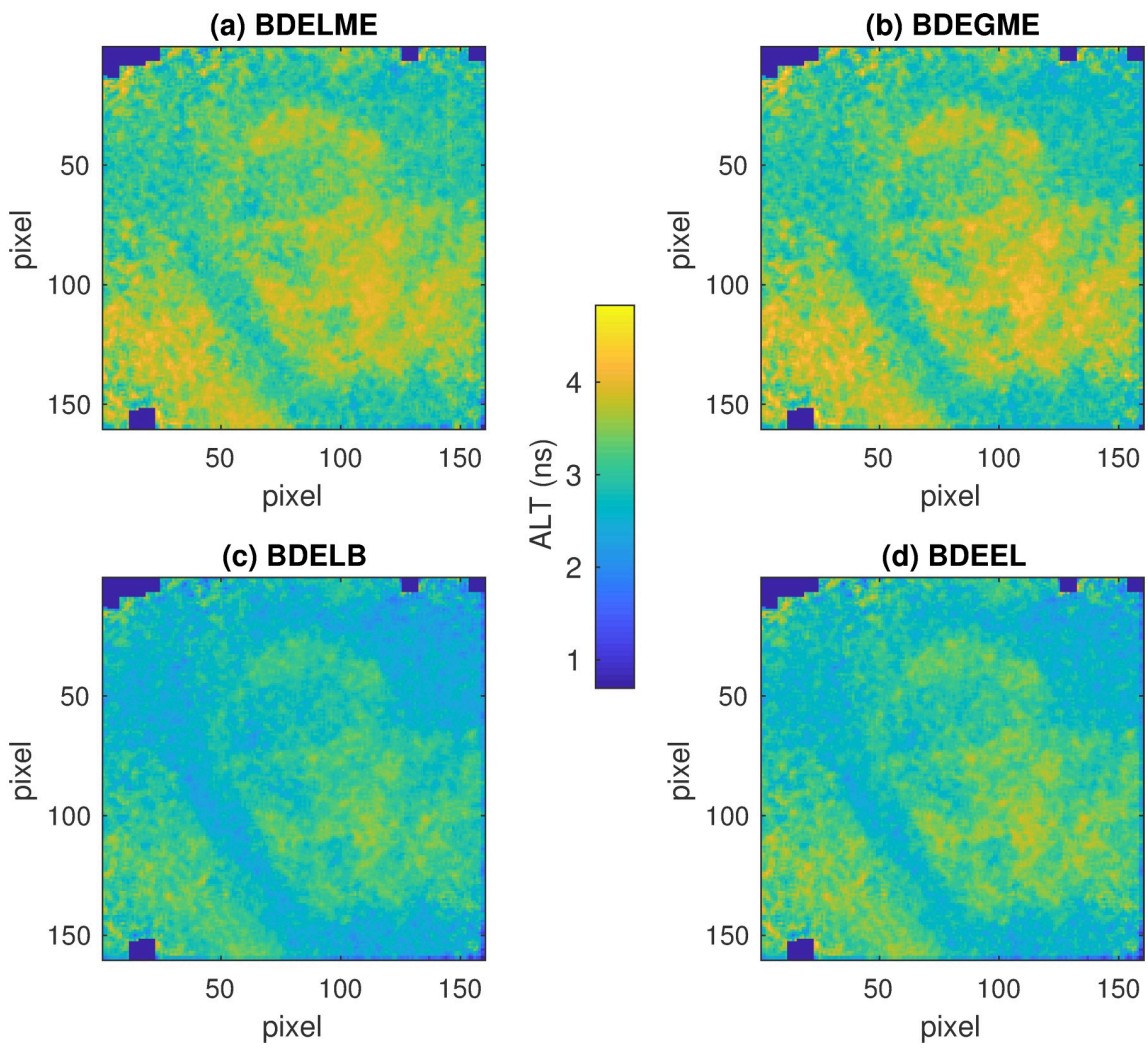

**Fig 12. Average lifetime maps for lesion sample No. 82 in 3rd spectral channel: (a) BDELME, (b) BDEGME, (c) BDELB, and (d) BDEEL.**

subplots illustrate the same morphological patterns with just small differences in the minimum and maximum ALTs. Finally, Fig 13 presents the B&A plots for all pairs of ALT estimations, and the corresponding correlation coefficients. In all cases, the correlation coefficients are larger than 0.94, which highlights high consistency among all BDE techniques.

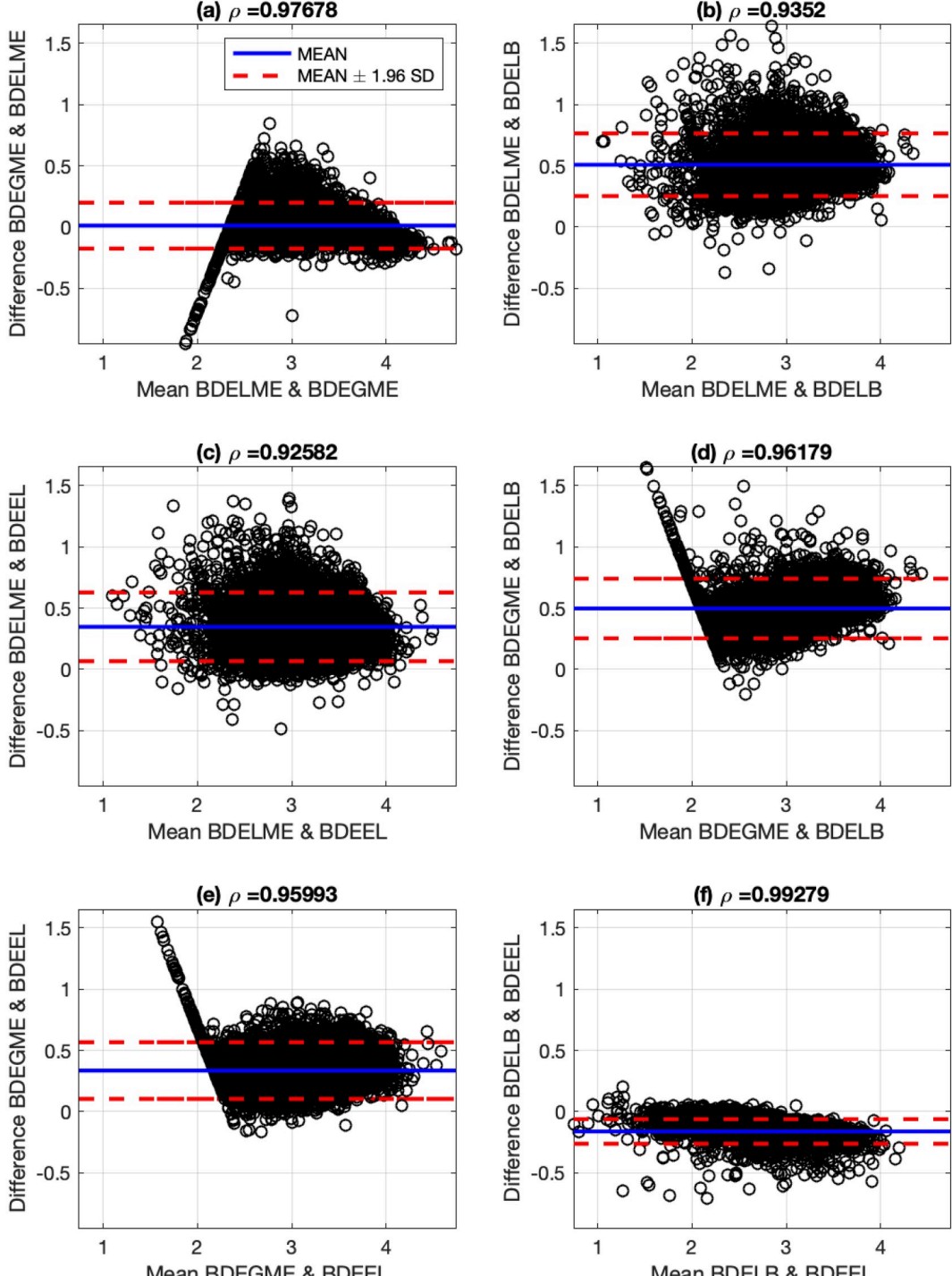

**Fig 13. Bland-Altman plots for estimated average lifetimes for lesion sample No. 82 in 3rd spectral channel and correlation coefficients: (a) BDELME vs BDEGME, (b) BDELME vs BDELB, (c) BDELME vs BDEEL, (d) BDEGME vs BDELB, (e) BDEGME vs BDEEL, and (f) BDELB vs BDEEL.**

## Conclusions

In this work, we introduced two new BDE algorithms based on a linear combination of multi-exponential functions for the FluoIRs modeling. Hence, the proposed algorithms estimate the FluoIRs in all the spatial points of the dataset and, simultaneously, the InstR for the fluorescence excitation. The InstR is assumed with a free-form and a sparcity condition. The local perspective of the BDE methodology (BDELME) assumes that the characteristic parameters of the exponential functions (time constants and scaling coefficients) are estimated for each spatial point of the dataset, i.e. pixel-by-pixel. On the other hand, for the global perspective, the exponential functions are assumed common to all the points in the dataset, and just their scaling coefficients are updated for each spatial point. By using a convolution modeling between the FluoIRs and InstR for the measured FluoDs, the time samples of the InstR and the scaling coefficients of the exponential functions exhibit a linear dependence in the observation model, but for the exponentials time constants, the dependence is nonlinear. To overcome the nonlinear interaction on the decision variables, an alternating least squares (ALS) methodology iteratively solves both estimation problems based on non-negative and constrained optimizations. The validation stage considered synthetic datasets at different noise types and levels, and a comparison with the standard deconvolution techniques: DELME and DEGME, as well as, two more BDE methodologies in the state of the art: BDELB and BDEEL. In the validation with experimental datasets, fluorescent dyes and oral tissue samples were considered. Our results show that BDELME and BDEGME reached the fastest convergence with the best compromise in FluoIRs and InstR estimation errors compared to BDELB and BDEEL. Also, the estimation performance of BDELME and BDEGME was consistent with the standard deconvolution techniques that assume the InstR is available: DELME and DEGME. For future work, we will develop parallel implementations of all the BDE methodologies to reduce computational time, as well as design a more comprehensive comparison with diverse tissue samples.

## Supporting information

**S1 File.**
(PDF)

## Author Contributions

**Conceptualization:** Daniel U. Campos-Delgado, Ricardo Salinas-Martinez, Javier A. Jo.

**Data curation:** Elvis Duran.

**Funding acquisition:** Daniel U. Campos-Delgado, Javier A. Jo.

**Investigation:** Ricardo Salinas-Martinez.

**Methodology:** Daniel U. Campos-Delgado.

**Software:** Omar Gutierrez-Navarro, Ricardo Salinas-Martinez, Elvis Duran, Miguel J. Velazquez-Duran.

**Supervision:** Daniel U. Campos-Delgado, Javier A. Jo.

**Validation:** Daniel U. Campos-Delgado, Omar Gutierrez-Navarro, Aldo R. Mejia-Rodriguez, Javier A. Jo.

**Visualization:** Daniel U. Campos-Delgado, Miguel J. Velazquez-Duran.

**Writing – original draft:** Daniel U. Campos-Delgado, Ricardo Salinas-Martinez.

**Writing – review & editing:** Daniel U. Campos-Delgado, Omar Gutierrez-Navarro, Ricardo Salinas-Martinez, Aldo R. Mejia-Rodriguez, Miguel J. Velazquez-Duran, Javier A. Jo.

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
