## [Decision Letter · Decision Letter 0]

7 Dec 2020

PONE-D-20-29880

Blind deconvolution estimation by multi-exponential models and alternated least squares approximations

PLOS ONE

Dear Dr. Campos-Delgado,

Thank you for submitting your manuscript to PLOS ONE. After careful consideration, we feel that it has merit but does not fully meet PLOS ONE’s publication criteria as it currently stands. Therefore, we invite you to submit a revised version of the manuscript that addresses the points raised during the review process.

We look forward to receiving your revised manuscript.

Kind regards,

Hao Sun

Academic Editor

PLOS ONE

Journal Requirements:

2. Thank you for stating the following in your Competing Interests section: 'No'

Additional Editor Comments:

I agree with the reviewer's comments. Please make sure to address them in the revised version.

Reviewers' comments:

Reviewer's Responses to Questions

**Comments to the Author**

1. Is the manuscript technically sound, and do the data support the conclusions?

Reviewer #1: Yes

2. Has the statistical analysis been performed appropriately and rigorously? 

Reviewer #1: No

3. Have the authors made all data underlying the findings in their manuscript fully available?

Reviewer #1: Yes

4. Is the manuscript presented in an intelligible fashion and written in standard English?

Reviewer #1: Yes

5. Review Comments to the Author

Reviewer #1: This work proposed a new method for solving blind deconvolution estimation (BDE) problem in fluorescence imaging microscopy, based on multi-exponential models and alternating least squares. Blind deconvolution is a well studied problem recently, and the idea of alternating minimization idea is not quite new. The reviewer does not find much technical novelty, but an application to a new problem. The work is well-organized overall, while the presentation can be improved. Below are some comments:

The abbreviations for fluorescence impulse response (FIR) and instrument response (IR) might be a bit confusing for readers with signal processing background. In signal processing, FIR and IIR are typically for finite impulse response and infinite impulse response, respectively. The authors should consider how to differ from the traditional abbreviations, otherwise readers might got confused with the meaning of FIR and IR at the first glance.

2. There are a couple of recent papers are related and might be cited:

[1] Joshua T Vogelstein, Adam M Packer, Tim A Machado, Tanya Sippy, Baktash Babadi, Rafael Yuste, Liam Paninski, Fast non-negative deconvolution for spike train inference from population calcium imaging

[2] Johannes Friedrich, Pengcheng Zhou, Liam Paninski, Fast online deconvolution of calcium imaging data

[3] Yenson Lau, Qing Qu, Han-wen Kuo, Pengcheng Zhou, Yuqian Zhang, John Wright. Short-and-Sparse Deconvolution – A Geometric Approach, ICLR’20.

In particular, the reviewer finds the proposed method is quite close to those of [1,2], though [1,2] considered a different application in Calcium imaging. In [1,2], the authors assumed some sparsity structures on FIR, while the proposed method does not. Some more comments and detailed comparison discussions are needed. In particular, the method might be improved by imposing sparsity structure on FIRs similarly, or other structures.

3. Since the problem is highly nonconvex, can the authors provide some convergence guarantees of the proposed optimization method? Some numerical and even theoretical justification is needed. Additionally, for convolution, the implementation could be faster via FFTs.

4. In numerical comparisons, the authors only compared the previous method by the authors themselves. Comparing with blind deconvolution methods such as those mentioned [1-3], or other generic methods are needed.

6. PLOS authors have the option to publish the peer review history of their article (what does this mean?). If published, this will include your full peer review and any attached files.

Reviewer #1: **Yes: **Qing Qu

---

## [Author Response · Author response to Decision Letter 0]

23 Jan 2021

First, the authors would like to thank the reviewer and the Editor for their valuable comments. The main contribution on this new version is the inclusion of a deeper critical perspective to the manuscript. We really think that such comments helped to improve the quality of the manuscript. 

%%%%%%%%%%%%%%%%%%%%%

Journal Requirements:

RESPONSE

We have carefully revised the PLOS ONE template and we are following all the style requirements in this new version. For the manuscript, we have used the LaTeX template, so all the styles for the heading and fonts are followed.

%%%%%%%%%%

2. Thank you for stating the following in your Competing Interests section: 'No'

RESPONSE

We have included in the cover letter for this new submission the statement "The authors have declared that no competing interests exist." 

%%%%%%%%%%

Additional Editor Comments:

I agree with the reviewer's comments. Please make sure to address them in the revised version.

RESPONSE

We sincerely thank the Editor for the evaluation of our manuscript. In the following, we have included a precise response to each reviewer comment. 

%%%%%%%%%%%%%%%%%%%%%%%

Reviewer #1: Comments to the Author 

This work proposed a new method for solving blind deconvolution estimation (BDE) problem in fluorescence imaging microscopy, based on multi-exponential models and alternating least squares. Blind deconvolution is a well studied problem recently, and the idea of alternating minimization idea is not quite new. The reviewer does not find much technical novelty, but an application to a new problem. The work is well-organized overall, while the presentation can be improved. Below are some comments: 

1. The abbreviations for fluorescence impulse response (FIR) and instrument response (IR) might be a bit confusing for readers with signal processing background. In signal processing, FIR and IIR are typically for finite impulse response and infinite impulse response, respectively. The authors should consider how to differ from the traditional abbreviations, otherwise readers might got confused with the meaning of FIR and IR at the first glance.

RESPONSE

We acknowledge this important remark by the reviewer, so we have updated the abbreviations for fluorescence impulse response and instrument response to FluoIR and InstR, respectively to avoid a possible confusion to the reader. For consistency, the abbreviation of fluorescence decay was also updated to FluoD in the whole manuscript (see page 20).

%%%%%%%%%%

2. There are a couple of recent papers are related and might be cited:

[1] Joshua T Vogelstein, Adam M Packer, Tim A Machado, Tanya Sippy, Baktash Babadi, Rafael Yuste, Liam Paninski, Fast non-negative deconvolution for spike train inference from population calcium imaging

[2] Johannes Friedrich, Pengcheng Zhou, Liam Paninski, Fast online deconvolution of calcium imaging data

[3] Yenson Lau, Qing Qu, Han-wen Kuo, Pengcheng Zhou, Yuqian Zhang, John Wright. Short-and-Sparse Deconvolution – A Geometric Approach, ICLR’20.

In particular, the reviewer finds the proposed method is quite close to those of [1,2], though [1,2] considered a different application in Calcium imaging. In [1,2], the authors assumed some sparsity structures on FIR, while the proposed method does not. Some more comments and detailed comparison discussions are needed. In particular, the method might be improved by imposing sparsity structure on FIRs similarly, or other structures.

RESPONSE

We thank deeply the reviewer for bringing to our attention these three relevant works [1,2,3] related to blind deconvolution estimation (BDE). We reviewed carefully the three previous contributions and we detected key differences with our BDE formulation. Contrary to [1,2], in our approach, the excitation signal does not have a spike train shape, and our observation model relies on a convolution with a multi-exponential kernel, compared to the autoregressive structure in the references. Furthermore, the main goal in our work is to reconstruct the fluorescence impulse response in each pixel of the FLIM measurement, nor to identify the spike train of the excitation signal, as in [1,2]. In addition, our formulation considers a free-form for the excitation signal, which is common to all pixels in the FLIM image, but with some sparsity condition as well. Hence, for FLIM datasets, the excitation signal is one narrow pulse without repetitions. Meanwhile, the fluorescence impulse responses have a smooth monotonic exponential decay over the whole measurement interval, so we cannot assume for them a sparsity condition, as suggested by the reviewer. With respect to [3], our formulation does not assume a short kernel and a sparse activation map. In addition, our multi-exponential kernel is not restricted to a sphere constraint, as in [3]. Also, the extension in [3] to convolutional dictionary learning considers multiple unknown kernels/motifs, i.e., multiple input signals, which is not consistent with the studied BDE formulation. Nonetheless, references [1,2,3] ([33], [34] and [35] in our manuscript) are described in the introduction and the main differences to our formulation are also discussed there (see page 2).

%%%%%%%%%%

3. Since the problem is highly nonconvex, can the authors provide some convergence guarantees of the proposed optimization method? Some numerical and even theoretical justification is needed. Additionally, for convolution, the implementation could be faster via FFTs.

RESPONSE

We acknowledge the detailed and precise evaluation of our work by the reviewer. Since in our formulation for FLIM datasets, we assume a narrow pulse without repetitions as excitation, and each measured fluorescence decay will exhibit a sharp increase to its peak value, followed by a monotonic decrease. Moreover, all the measurements are scaled to sum-to-one, and also the instrument response is limited to sum-to-one. As a result, the scaled shift symmetry described in [3] will not hold in our BDE formulation. In addition, at each stage of the alternated least squares (ALS) scheme in the local and global approaches, a quadratic approximation problem is solved by either a non-linear least squares or linear least squares. So, at each stage, the estimation error is reduced or at least maintained. Consequently, convergence is guaranteed in the iterative scheme, but only to a local minimum. For this reason, the initialization based on processing the FLIM dataset is a crucial step in our formulation to obtain meaningful results. We include this discussion in third section, while describing the alternated least squares structure (see page 7). 

In this new version, we have added a numerical convergence evaluation of the BDE schemes (BDELME and BDEGME) with synthetic FLIM datasets for different noise levels (SNR,PSNR) ∈{(10,40),(15,45),(20,50),(25,55)} dB, and orders in the multi-exponential model of the synthetic impulse response (2nd, 3rd and 4th). The stopping threshold is set to ϵ_3=1×10^(-3) . The resulting normalized metric in the alternated least squares iterations is described below in Fig. 4 of the new version, where it is observed that in either BDE scheme, the convergence is always monotonic for any noise combination and model order, and just its final steady-state value depends on the noise level. From this figure, we conclude that the local approach (BDELME) converges slower than the global scheme (BDEGME), which could be intuitively expected, since BDELME has roughly twice the free parameters to optimize. For BDELME, the most drastic reductions are achieved in the first five iterations, and for BDEGME, in the first two. In this way, our previous convergence analysis is validated numerically. This previous analysis and discussion is included in page 12.

Now, with respect to the suggestion of implementing the convolution via FFTs, we have chosen to use a time-domain perspective to allow a direct optimization by nonlinear and linear least squares of the parameters in the impulse and instrument responses, nor for the computation of the estimated fluorescence decays. Also, our formulation considers a linear convolution for the observation model. Nonetheless, we thank the reviewer for this interesting suggestion.

%%%%%%%%%%

4. In numerical comparisons, the authors only compared the previous method by the authors themselves. Comparing with blind deconvolution methods such as those mentioned [1-3], or other generic methods are needed.

RESPONSE

We thank this important observation by the reviewer. Nonetheless, the numerical comparison must be under the same mathematical formulation for the deconvolution process. Hence, as we argued in response 2), the performance comparisons with [1], [2], and [3] are not possible, since they address a different estimation problem. In the “Synthetic and experimental validation” section (see pages 10 to 16), we are comparing the performance of our proposed BDE schemes, BDELME and BDEGME, first to two standard deconvolution methodologies that assume known a priori the instrument response: DELME and DEGME (Pelet et al, 2004 & Warren et al., 2013) [17,18]. These two strategies are the common standards in the numerical analysis of FLIM datasets. Next, we use two previous BDE proposals of our same research group: BDELB and BDEEL. In this way, the proposed methodologies, BDELME and BDEGME, are compared against four schemes from the literature, where their advantages are illustrated in synthetic and experimental datasets (see page 11). We hope that the reviewer could identify our effort to compare our proposals, and to evaluate their performance without bias. 

[17] Pelet S, Previte M, Laiho L, So P. A fast global fitting algorithm for fluorescence lifetime imaging microscopy based on image 

segmentation. Biophysical Journal. 2004;87(4):2807–2817. doi:10.1529/biophysj.104.045492.

[18] Warren SC, Margineanu A, Alibhai D, Kelly DJ, Talbot C, Alexandrov Y, et al. Rapid global fitting of large fluorescence lifetime imaging microscopy datasets. PLoS One. 2013;8(8):e70687. doi:10.1371/journal.pone.0070687.

---

## [Decision Letter · Decision Letter 1]

24 Feb 2021

Blind deconvolution estimation by multi-exponential models and alternated least squares approximations:  free-form and sparse approach

PONE-D-20-29880R1

Dear Dr. Campos-Delgado,

We’re pleased to inform you that your manuscript has been judged scientifically suitable for publication and will be formally accepted for publication once it meets all outstanding technical requirements.

Kind regards,

Hao Sun

Academic Editor

PLOS ONE

Reviewers' comments:

Reviewer's Responses to Questions

**Comments to the Author**

1. If the authors have adequately addressed your comments raised in a previous round of review and you feel that this manuscript is now acceptable for publication, you may indicate that here to bypass the “Comments to the Author” section, enter your conflict of interest statement in the “Confidential to Editor” section, and submit your "Accept" recommendation.

Reviewer #1: All comments have been addressed

2. Is the manuscript technically sound, and do the data support the conclusions?

Reviewer #1: Yes

3. Has the statistical analysis been performed appropriately and rigorously? 

Reviewer #1: Yes

4. Have the authors made all data underlying the findings in their manuscript fully available?

Reviewer #1: Yes

5. Is the manuscript presented in an intelligible fashion and written in standard English?

Reviewer #1: Yes

6. Review Comments to the Author

Reviewer #1: The authors have successfully addressed all my concerns, that I recommend for acceptance in the current form.

7. PLOS authors have the option to publish the peer review history of their article (what does this mean?). If published, this will include your full peer review and any attached files.

Reviewer #1: No

---

## [Editor Report · Acceptance letter]

8 Mar 2021

PONE-D-20-29880R1 

Blind deconvolution estimation by multi-exponential models and alternated least squares approximations: free-form and sparse approach  

Dear Dr. Campos-Delgado:

I'm pleased to inform you that your manuscript has been deemed suitable for publication in PLOS ONE. Congratulations! Your manuscript is now with our production department. 

Kind regards, 

on behalf of

Professor Hao Sun 

Academic Editor

PLOS ONE